## PROCEEDINGS A

# Research

mechanical engineering

rheology, particulate composite, particulate suspension, capillary flow, additive manufacturing

**Author for correspondence:**
Paul S. Krueger
e-mail: pkrueger@lyle.smu.edu

# Rheology of particulate suspensions with non-Newtonian fluids in capillaries

## Bin Xia and Paul S. Krueger

Department of Mechanical Engineering, Southern Methodist University, Dallas, TX 75205, USA

BX, 0000-0002-1973-1159; PSK, 0000-0001-8697-6722

Particulate suspensions occur in situations from blood flow to slurries in drilling applications. Existing investigations of these suspensions generally concentrate on the impact of particle volume fraction for suspensions in Newtonian fluids under free-flow conditions. Recently, particulate-polymer composites have been used in additive manufacturing (AM). Here, the polymer becomes a shear-thinning non-Newtonian fluid during extrusion, creating a particulate suspension. Motivated by the challenges in AM of particulate composites, this study investigates the rheology of suspensions of micrometre-sized particles in shear-thinning silicone while extruded through AM-scaled nozzles (millimetre-scale diameters). The suspensions were observed to follow a power-law behaviour and their rheology was investigated through the measured flow consistency ($K$) and behaviour ($n$) indices. The impact of the particle volume fraction ($\phi$) and the ratio ($\omega$) of the capillary inside diameter to the particle diameter on both indices were measured. $n$ was found to be only impacted by the suspension fluid type and $\phi$. $K$ was found to be constant at large $\omega$, but decreased and then increased to infinity with $\omega$ decreasing. Based on its behaviour, $K$ was categorized into two conditions and analysed separately with semi-empirical models. The impact of particle size distribution was also investigated.

## 1. Introduction

Particulate composites are the composites made of particles embedded in a matrix material. They are widely used in multi-functional additive manufacturing (AM),

**Table 1.** Significant viscosity models for particulate suspensions[a]

| authors | type | equation |
|---|---|---|
| Einstein | linear | $\mu = \mu_0(1 + 2.5\phi)$ |
| Guth, Eugene & Simha | polynomial | $\mu = \mu_0(1 + 2.5\phi + 14.1\phi^2)$ |
| Vand | exponential | $\mu = \mu_0 \exp((2.5\phi + 2.7\phi^2)/(1 - 0.609\phi))$ |
| Mooney | exponential | $\mu = \mu_0 \exp(2.5\phi/(1 + k\phi))$ |
| Simha | polynomial | $\mu = \mu_0(1 + 1.5\phi(1 + (1 + 25/4f^3)\cdots))$ |
| Brinkman | power law | $\mu = \mu_0(1 - \phi)^{-[\mu]}$ |
| Krieger & Dougherty | power law | $\mu = \mu_0(1 - (\phi/\phi_M))^{-[\mu]\phi_M}$ |
| Ford | polynomial | $\mu = \mu_0(1 + 2.5\phi + 11\phi^5 - 11.5\phi^7)$ |
| Thomas | mixed | $\mu = \mu_0(1 + 2.5\phi + 10.5\phi^2 + 0.00273\, e^{16.6\phi})$ |
| Bournonville & Nzihou | power law | $\mu = \mu_0(1 + (D/\dot{\gamma}^E)(\phi_v/\phi_M/(1 - \phi_v/\phi_M)))^G$ |
| Senapati | power law | $\mu = \mu_0(10C_u/d_{50})(1 + ([\mu]/\dot{\gamma}^{0.4})(\phi/(\phi_M - \phi)))^{3.5}$ |
| Blissett | mixed | $\mu = \mu_0(1 - (\phi/\phi_M))^{-[\mu]\phi_M} + m(\phi)\gamma^{n(\phi)-1}$ |

[a]In the table, $\phi$ is the particle volume fraction, $[\mu] = \lim_{\phi \to 0}((\mu - \mu_0)/\phi\mu_0)$ is the intrinsic viscosity, $\phi_M$ is the maximum particle volume fraction (capacity of the suspension fluid for accepting particles) and the other parameters are described in the text.

particularly with polymer matrix materials, as the particles can provide different functionalities while the matrix material maintains the overall extrudability of the composite. In experiments with extrusion-based AM of particulate composites [1], it was discovered that the extruding nozzles may be blocked by the particles and the extrusion force varied dramatically with different extruding conditions. As the composites can be regarded as micrometre-scale particles suspended in shear-thinning fluid under extrusion conditions, they can be treated as a particulate suspension flowing through a millimetre-scale capillary. Hence, the rheology of the suspension governs its extrusion behaviour, which is important for extrusion deposition in AM [2,3]. Motivated by an interest in improving the AM of particulate composites, this work seeks to investigate and model the rheology of particulate suspensions based on shear-thinning fluids flowing in capillaries.

Existing background in this area is sparse as there is currently no specific model describing the rheology of particulate suspensions in capillaries. Most investigations of particulate suspension rheology focus on the particles suspended in Newtonian fluids and the effect of particle volume fraction on fluid viscosity under free-flowing conditions (negligible impact of boundaries on particle behaviour). A summary of various models developed to describe the behaviour of the suspension viscosity under these conditions is presented in table 1. The first viscosity model was developed by Einstein in 1906 [4] (as cited in [5]) and has a linear dependence of viscosity on the particle volume fraction, $\phi$. The linear equation only considers the no-slip boundary condition over the particle sphere in purely laminar flow. The allowable particle diameter ($d$) and volume fraction range are very limited. Since Einstein, various methods have been employed to increase the accuracy and applicable range of these models.

In 1936, Guth *et al.* [6] (as cited in [5]) increased the range of applicability up to $\phi = 0.2$ using a second-order polynomial expression. The coefficient of the second-order term was determined from a method of successive reflection, which assumed that the disturbance of flow around a first sphere was compensated by an additional flow around a second sphere to fulfil the continuity equation and no-slip boundary condition at the sphere's surface. Simha [5] increased the accuracy further in 1952 by including more terms and adding the semi-empirical parameter $f$ to fit dilute suspensions. Ford [7] worked on low/moderate concentration suspensions in 1960 and developed a seventh-order polynomial model.

Taking a different approach, in 1948, Vand [8] derived an exponential function from the Navier–Stokes equations considering the effect of adding an incremental volume fraction of

spheres, d$\phi$, and accounting for the interactions of particles using the same method of successive reflection as Guth *et al.* [6]. In 1951, Mooney [9] developed a similar model by considering two successive additions of monodisperse spheres to a pure fluid, accounting for possible hydrodynamic interactions and the mutual crowding effects of the two-sphere populations on each other using an experimentally determined parameter $k$ (crowding factor) where $1.35 < k < 1.91$.

In 1952, Brinkman [10] developed a power law model; however, the original model is limited to the special case of infinite polydispersity (meaning the maximum particle volume fraction, $\phi_M$, is equal to 1). Krieger & Dougherty [11] improved this model by limiting the maximum particle fraction using Mooney's concept of a 'crowding factor'. In 1984, Wildemuth [12] introduced a parameter considering shear-rate dependent maximum volume fraction. (When the particles are not spherical, the particle orientation is impacted by the shear, resulting in a different maximum volume fraction.) In 2002, Bournonville & Nzihou [13] introduced three adjustable empirical constants, $D$, $E$ and $G$ to allow for the applicability at very high ($\sim 10^6 \, \mathrm{s}^{-1}$) and low ($\sim 0.01 \, \mathrm{s}^{-1}$) shear rate. In 2009, Senapati [14] introduced two adjustable empirical constants $C_u$ and $d_{50}$ to account for the separate effects of median particle diameter and particle diameter distribution. Additional approaches involve empirical modifications such as the addition of an exponential term to a polynomial as modelled by Thomas [15] (but with no theoretical explanation given for the additional term) and the addition of a second term with two empirical constants to the power-law model of Krieger & Dougherty [11] by Blissett [16].

All models described in table 1 are based on suspensions in Newtonian fluids, which means the adjustable parameters are determined by the particles. However, when the fluid used in the suspension is non-Newtonian, the viscosity can be affected by both the fluid properties and the particles, and these may interact with each other. The coupled factors may add additional complexity.

A few recent investigations consider particulate suspensions in shear-thinning polymer melts [17–22]. Kataoka *et al.* [19–21] applied Mooney's model to polymer melt suspensions to calculate the relative viscosity of the suspension and suspension fluid under the same shear stress and modified it with another adjustable parameter to apply to polymer melts containing a suspension of short fibres [18]. The applicable range of $\phi$ in their model was limited (only $\phi = 10\%$ was investigated) and a rotational rheometer was used for the empirical data, which may have provided unreliable results as described below.

The observations of suspensions in both Newtonian and non-Newtonian fluids relate that with higher $\phi$, the materials change from a fluid-like to a solid-like state with an observable yield stress, which is a phase change called the jamming transition [23]. The jamming transition occurs at a characteristic volume fraction ($\phi_M$) [24], which in general may depend on the nature of the particles and the flow state (e.g. shear rate) of the suspension [25–27]. As $\phi \to \phi_M$, lubrication layers between particles begin disappearing and the number of frictional contacts per particle increases [25]. At $\phi_M$, the suspension reaches the 'maximum packing fraction possible for a given suspension composition and packing arrangement' [24].

Much of the work on jamming behaviour focuses on modelling the mechanics of the jamming process and on determining the jamming transition using generic shear flows [25–29]. In AM applications, a significant factor impacting jamming includes confinement effects from the capillary inner diameter ($D$) in relation to the particle mean diameter ($d$) and its distribution, but this has not been considered to date. Additionally, most of the work uses particle-induced shear thickening material (particle suspensions in a Newtonian fluid that become shear thickening after adding particles) [28,30,31] or viscoelastic materials in a squeeze geometry [32], but most of the polymer materials used in AM are shear thinning.

Prior work in AM has also considered the flow behaviour of particulate composites. Some studies have investigated the rheology of carbon fibre (CF) reinforced polymer composites showing general shear-thinning behaviour and that the addition of CF can increase the shear-thinning characteristics of the polymer melt [33,34], but the range of $\phi$ investigated was limited and confinement effects were not considered. Wang and Smith [35,36] used computational

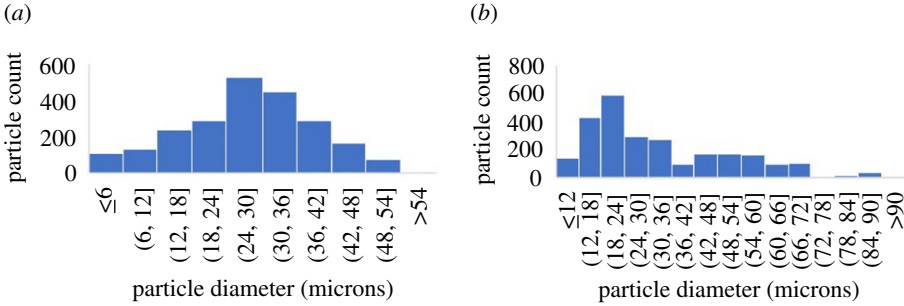

**Figure 1.** Particle diameter distributions for (*a*) FG 22 and (*b*) A3000. (Online version in colour.)

methods to simulate the flow behaviour of fibre-based polymer composites inside a nozzle, including model rheology effects. The emphasis in these studies, however, was on fibre orientation following printing and the resulting mechanical properties of the solid printed material.

The objective of this investigation is to investigate the rheology of particulate suspensions in shear-thinning non-Newtonian fluids, including the jamming effects associated with extrusion through confined channels (capillaries), and provide a model for describing the observed behaviour. Such information will be helpful in understanding the behaviour of particulate composites under different processing conditions, particularly those relevant for AM where a range of particle loading may be used and the confining effects of the small extrusion nozzles can make extruding these materials challenging.

In this work, §2 presents the design of the rheometer used to investigate the particulate suspensions and the range of conditions investigated. Section 3 presents the measured rheology of the suspensions in terms of the flow consistency index ($K$) and flow behaviour index ($n$) for the observed power-law behaviour of the suspensions. Section 4 develops models describing measured trends in $K$ and $n$. Section 5 presents the conclusions.

## 2. Experiment design and set-up

The suspensions formulated for this investigation used Momentive UV-Electro 225-1 Base silicone ($K = 93 \, \text{Pa}^n \cdot s$ and $n = 0.816$ [−], referred to as UV 225-1 throughout) and a corn syrup mixture ($K = 62 \, \text{Pa}^n \cdot s$ and $n = 1.00$ [−], 95% ADM corn syrup 42/43 and 5% water (volume fraction), referred to as 'ADM' throughout) as the suspension fluids. The UV 225-1 is a shear-thinning liquid that was used in place of polymer melts to avoid dealing with high temperatures and the corn syrup mixture provided a Newtonian suspension fluid for comparison purposes. Two types of particles were used in the suspensions: Fibre Glast Microspheres 22 ($d = 0.0420 \, \text{mm}$, referred to as FG22 throughout) and Spheriglass® Solid Glass Microspheres A3000 ($d = 0.0299 \, \text{mm}$, referred to as A3000 throughout). The mean diameters and size distribution of both particles were measured by imaging over 2000 of each type with a microscope (OLYMPUS BX60F-3 10 × 20) and extracting the particle sizes from the images using MATLAB (with an accuracy of ±0.0014 mm). Their particle diameter distributions are shown in figure 1.

Suspensions were prepared by mixing the particles and suspension fluids at the desired weight ratio with an orbital mixer (Flacktek Speedmixer DAC 150.1 FVZ-K) for around 5 min at a speed of 3000 r.p.m. (to make the suspension uniform and eliminate air bubbles). The weights of the materials were measured with a scale (Cole-Parmer Symmetry UX-20000-34). The required weight of each component was calculated according to their density and required particle volume fraction (accurate to within 0.5%).

To quantify the rheology of the prepared suspensions, a capillary rheometer was designed and built as illustrated in figure 2. A capillary rheometer was used as opposed to other rheometer

*(a)*  *(b)*

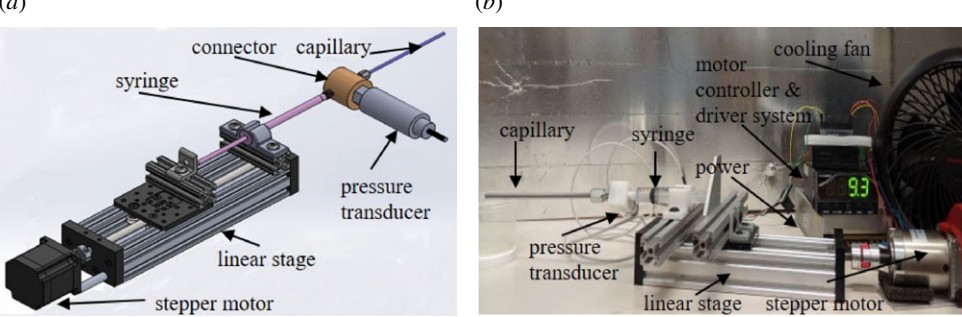

**Figure 2.** (*a*) Rheometer schematic and (*b*) the actual system. (Online version in colour.)

configurations as particles may migrate in a rotary rheometer (affecting the local material properties) and the capillaries introduce a confinement effect similar to AM extrusion nozzles that can be investigated by using capillaries with different $D$. A motorized linear stage (OpenBuild C-Beam Linear Actuator Bundle and NEMA 23HS22-2804S-PG47 Stepper) with a linear accuracy of 0.091 mm was used to actuate the syringes and drive the flow. The capillaries were seamless steel tubing with $D$ of 0.3302, 0.4064, 0.6096, 0.8382, 1.36, 1.73, 2.01, 3.05, 3.861 and 4.572 mm and different lengths. Omega Engineering PX61V1-1KGI (accuracy of 690 Pa) and PX61v1-100GI (accuracy of 69 Pa) pressure transducers were used for measuring the pressure required to drive the flow through the capillaries. Within the rheometer, the flow rate $Q$ was controlled to keep the wall shear rate of the flow in the range of 0.5–50 $\text{s}^{-1}$ (see equations (2.1)–(2.3)) using the motorized stage. When the flow arrived at steady state, the pressure drop in the capillary from the pressure transducer was recorded.

To avoid pressure measurement inaccuracy because of the contraction and developing region at the capillary entrance, tests were done on the capillaries with two different lengths for the same $D$ as illustrated in figure 3. Lengths of the capillaries were determined to guarantee the flow in the region $L$ was (hydrodynamically) fully developed and laminar based on the related theories [37]. (The particle distribution remained uniform throughout as will be discussed in §4b, so no entrance length for this property was considered.) Then the pressure drop $\Delta P$ in the region $L$ was obtained from the difference in pressure drop for the two different length tubes (with the same radius $R = D/2$) at the same flow rate. By testing with different flow rates, the wall shear stress $\tau_w$, wall shear rate $\gamma_w$ and effective viscosity $\mu$ were calculated from the following results [38]:

$$\tau_w = \frac{R}{2L}\Delta P, \tag{2.1}$$

$$n = \frac{d(\ln(Q/\pi R^3))}{d(\ln(\tau_w))}, \tag{2.2}$$

$$\dot{\gamma}_w = \frac{4Q}{\pi R^3}\left(\frac{1}{4}n + \frac{3}{4}\right) \tag{2.3}$$

and

$$\mu = \frac{\tau_w}{\dot{\gamma}_w}. \tag{2.4}$$

For the suspensions tested, it was found that $n$ in equation (2.2) was constant for the same suspension in a capillary with the same $D$. Thus, it was concluded that the suspensions follow a power law model in which the shear stress can be described as

$$\tau = K\dot{\gamma}^n. \tag{2.5}$$

Here, $K$ is the flow consistency index and $n$ is the flow behaviour index. $\tau$ and $\dot{\gamma}$ are the shear stress and rate.

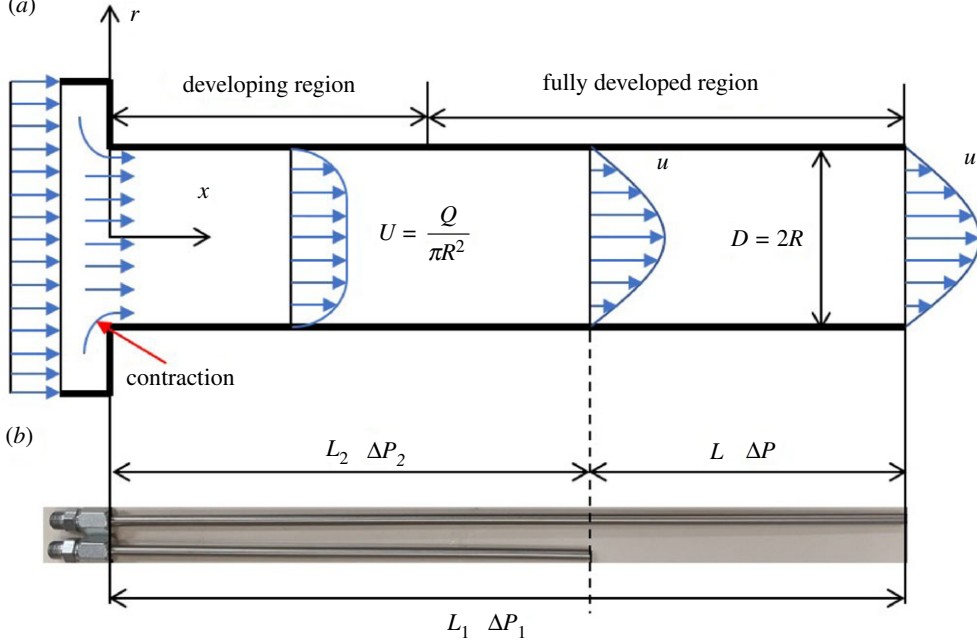

**Figure 3.** (*a*) Schematic diagram and (*b*) image of the capillaries with length $L_1$ and $L_2$. (Online version in colour.)

To quantify the impacts of particle volume fraction $\phi$, particle mean diameter $d$ and capillary inside diameter $D$ on the particulate suspension rheology in the capillary, the flow consistency index $K$ and flow behaviour index $n$ were obtained for the suspensions made of UV 225-1 and FG 22/A3000 ($\phi = 10\%, 20\%, 30\%, 40\%$) separately. Suspensions with ADM and FG 22 ($\phi = 30\%$) were also tested for comparison.

## 3. Experiment results

Results for $K$ and $n$ as a function of $\omega = D/d$ for the $\phi$ investigated are shown in figure 4. Suspensions with different formulations are indicated by symbols with different colours and shapes. The error bars represent the standard deviation of the results over five tests at each condition.

The results in figure 4 show that the UV 225-1 suspensions exhibit non-Newtonian, shear-thinning behaviour ($n < 1$) for all $\phi$ investigated with a weak dependence of $n$ on $\phi$, while the ADM suspension exhibited Newtonian behaviour ($n \approx 1$). Interestingly, a similar shape in the trend for $K$ (dotted lines) was observed for all cases tested. The trend line shape stays the same with varied particle volume fraction, suspension fluid and particle diameter, but seems to be shifted and scaled depending on particle type and volume fraction. A sharp rise in $K$ was observed as $\omega$ decreased, corresponding to the approach to the jamming transition as the capillary diameter approached the particle mean diameter.

Despite the flow contraction entering the capillary tube in the rheometer (figure 2), there did not appear to be self-filtration due to intermittent jamming at the contraction as reported in Haw [39]. For example, UV 225-1 suspensions with A3000 particles for $\phi = 30\%$ at $\omega = 43.0$ and a volume flow rate of $0.1\,\mathrm{ml\,min}^{-1}$ gave mass flow rates of $0.150 \pm 0.001\,\mathrm{g\,min}^{-1}$ leaving the capillary tube. Based on the density of the suspension fluid and particles, the expected mass flow rate was $0.150\,\mathrm{g\,min}^{-1}$, indicating the suspension fluid and particles were exiting the capillary in the same volume ratio at which they entered because the density of the particles was approximately 2.5 times larger than that of the fluid. The reason no self-filtration was observed

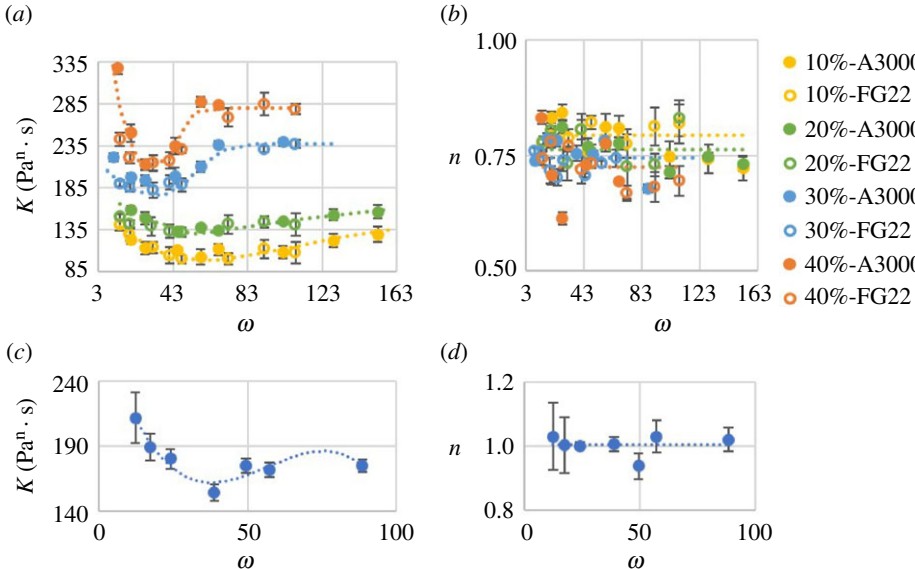

**Figure 4.** Flow consistency index $K(Pa^n \cdot s)$ and flow behaviour index $n$ versus ratio of capillary inside diameter to particle mean diameter $\omega$: (*a*) and (*b*) results for UV225-based suspensions with legend labels following the format: [particle volume fraction] – [particle type], (*c*) and (*d*) results for the ADM-based suspensions. (Online version in colour.)

could be because the critical $\phi$ to observe this behaviour was not reached, or because of differences in the suspension fluid, which in this case was both shear thinning and approximately $10^4$ times more viscous than the suspension fluid used in Haw [39]. Likely, a combination of these factors was at play.

# 4. Data analysis and modelling

In this section, models describing the behaviour of $K$ and $n$ in terms of $\omega$ and $\phi$ will be developed.

## (a) Flow behaviour index

As presented in figure 4*b*,*d*, $\omega$ had minimal effect on the flow behaviour index $n$, but it exhibited a weak dependence on $\phi$. Specifically, $n$ decreased with increasing $\phi$ for the shear-thinning suspension fluid investigated and was equal to 1 regardless of $\phi$ for the particulate suspension with a Newtonian suspension fluid, which is also the case for all the models in table 1. That is, over the range of conditions studied, for a suspension of particles in a Newtonian suspension fluid at fixed $\phi$, the suspension remained Newtonian, whereas the suspension became more shear thinning when particles were added to a shear-thinning suspension fluid.

A model for the flow behaviour index is described as

$$n = n_f(1 + a\phi), \tag{4.1}$$

where $n_f$ is the flow behaviour index of the suspension fluid and $a$ is an adjustable parameter. For the UV 225-1-based experiments, a least-square fit of equation (4.1) to the results gives $a = -0.23 \pm 0.01$ as shown in figure 5. For a Newtonian fluid, $a = 0$, and in general, $a$ may depend on the suspension fluid.

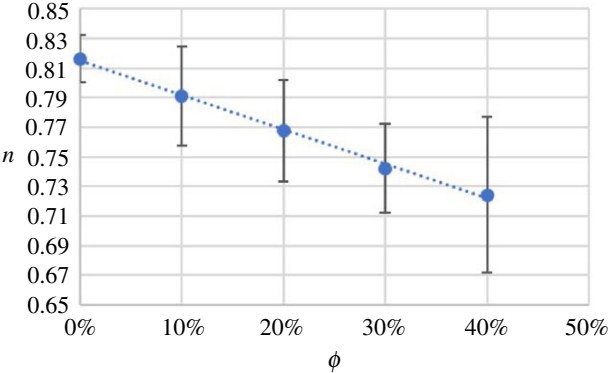

**Figure 5.** Flow behaviour index $n$ versus particle volume fraction $\phi$ (UV 225-1 suspension fluid). The error bars represent the standard deviations of all $n$ for each $\phi$, and the dotted line is the least-squares fit of equation (4.1) to the data. (Online version in colour.)

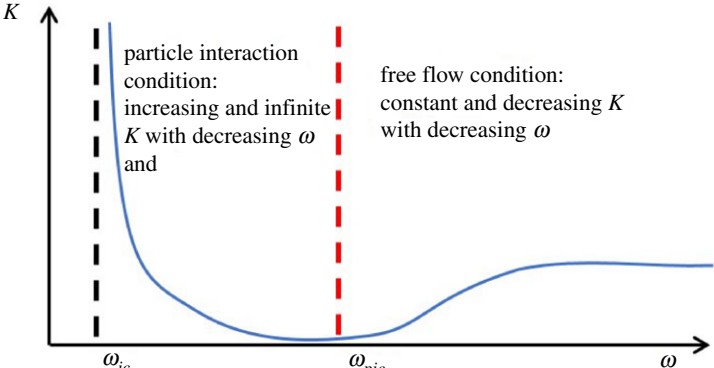

**Figure 6.** Categorization of different flow conditions observed for the dependence of $K$ on $\omega$ at fixed $\phi$. (Online version in colour.)

## (b) Flow consistency index

The flow consistency index shows a strong dependence on $\omega$. For large $\omega$, $K$ is approximately constant, but it initially decreases with decreasing $\omega$ and then increases as $\omega$ is decreased further toward zero. When $\omega$ is small enough, the capillary becomes blocked and the flow consistency index increases sharply towards infinity.

The general trend observed for $K$ at fixed $\phi$ is illustrated schematically in figure 6. The minimum in $K$ occurs at a boundary defined by $\omega_{pic}$, below which the rheology is dominated by increasing particle-to-particle and particle-to-wall frictional contacts as $\omega$ is decreased until sufficient contacts per particle are achieved to effect jamming at $\omega_{jc}$. The 'particle interaction' condition governs the flow for $\omega_{jc} < \omega < \omega_{pic}$, due to the strong contact interactions in this region. Conversely, the 'free flow' condition occurs for $\omega > \omega_{pic}$, where the rheology is dominated by an intervening lubrication layer of suspension fluid between the particles that allows for smooth suspension flow for the range of $\phi$ tested ($\phi < \phi_M$). This region includes both the constant $K$ behaviour as $\omega \to \infty$ and the dip in $K$ as $\omega$ decreases toward $\omega_{pic}$. The decrease in $K$ is not treated as separate behaviour as measurements [1] (discussed below) showed that in this region the particles were still uniformly distributed in the capillary cross section similar to larger $\omega$ behaviour.

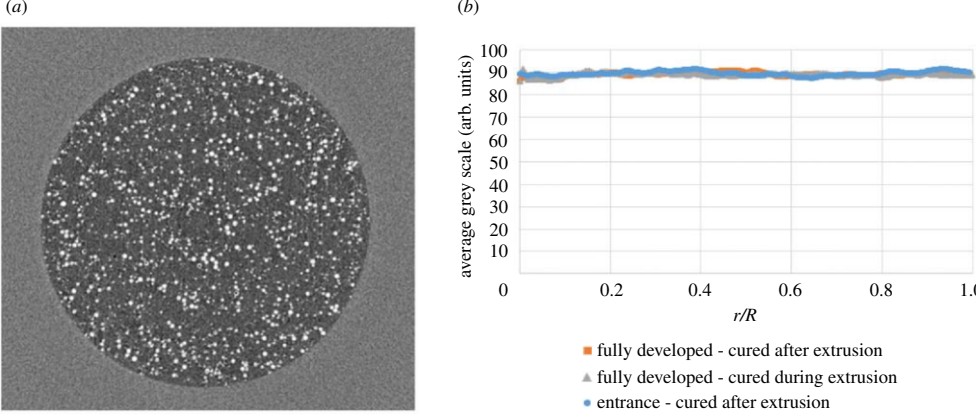

*(a)*    *(b)*

**Figure 7.** (*a*) Cross section of a micro-CT scan of a particulate suspension ($\phi = 10\%$), (*b*) Average grey scale of micro-CT scans versus radial position ($\phi = 10\%$).

The observed uniformity of the particle distribution in the capillary cross section deserves further discussion. Generally, the inhomogeneous shear experienced by the particles in these flows is known to induce a non-uniform particle distribution sufficiently far downstream in the tubes, even for shear-thinning suspension fluids [24,40,41]. This behaviour, however, was not observed in the present investigation. To confirm this, particle distributions inside the capillary at the entrance and (hydrodynamically) fully developed region were visualized via a micro-CT (SkyScan 1172 high-resolution desktop scanner at a resolution of 2.00 μm). For these measurements, samples were fabricated by extruding the suspension (with the addition of a curing agent) into transparent tubing ($D = 1.286$ mm) and then curing the silicone polymer with a UV light to create a solid sample that could be further analysed. Three samples were fabricated including samples cured during and after extrusion in the (hydrodynamically) fully developed region, and a sample cured after extrusion in the entrance region. A sample cross section obtained from the micro-CT scans is shown in figure 7*a* for $\phi = 10\%$ (UV 225-1 with A3000 particles). The lighter regions are the particles and the darker regions are the silicone. Hence, the radial particle density distribution can be obtained by calculating the average of grey scale of the sample at different radial positions for all axial cross sections in a given region. The result for this sample is shown in figure 7*b*, which shows the particle volume fraction is uniform in the tube cross section from the entrance to the fully developed region. Similar results were obtained for $\phi = 30\%$.

This difference between the present results and other studies showing particle migration and non-uniform particle distributions in particulate suspensions extruded through tubes is likely related to the differences in flow geometry and fluid properties. Tehrani [40] indicates the radial migration velocity ($V_r$) of particles in non-Newtonian fluids is determined by

$$V_r \propto d^2 We \frac{\partial \dot{\gamma}}{\partial r}, \tag{4.2}$$

where *We* is the Weissenberg number given by the ratio of the first normal stress difference to the shear stress ($N_1/\tau$). Then the entrance length for the particle distribution scales like $L_{ep} \sim UD/V_r$ where *U* is the average velocity in the tube. Using $\dot{\gamma} \sim U/D$ and $\partial \dot{\gamma}/\partial r \sim \dot{\gamma}/D$ gives

$$\frac{L_{ep}}{D} \sim \frac{\omega^2}{We}. \tag{4.3}$$

Considering a typical shear rate of $50\,\mathrm{s}^{-1}$, *We* for the 'fluid like' samples used in Tehrani [40] were in the range $2.2 \sim 36$, whereas $We \approx 0.47$ for UV 225-1 using correlations in Steller [42]. For the free-flow condition, $\omega > 30$, indicating that $L_{ep}/D$ for this investigation would be $\gtrsim 10^2$ times larger than that in Tehrani [40], leading to $L_{ep}/D > 1000$ for the present results. Higher $\phi$ would

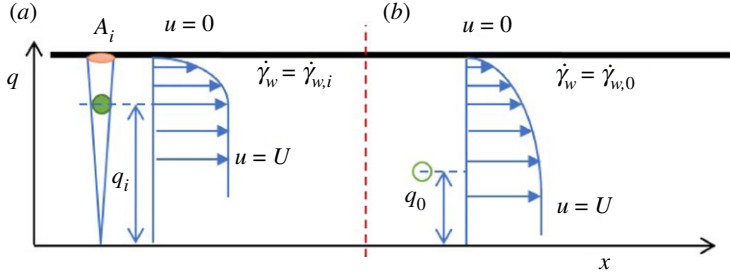

**Figure 8.** Schematic diagram of particles (a) inside and (b) outside the region where the particles may influence the wall shear rate. (Online version in colour.)

likely influence these predictions (Tehrani [40] used $\phi = 5\%$ and 12%), but increasing $\phi$ makes the suspensions investigated here more shear thinning (as noted above), which can produce more plug-like flow and slow particle migration (increase $L_{ep}$). So, while measurable particle migration might appear for extremely long capillary lengths, such behaviour would be unrealistic for the target application of AM (where flow paths tend to be relatively short) and uniform particle distributions will be assumed in the following.

Using the assumption of uniform particle distribution in the free-flow condition, mathematical models for the observed trends in $K$ are developed in the following based on the governing behaviour (free flow versus particle interaction) for each condition.

### (i) Model of the free-flow condition

The total shear force $F$ on the capillary wall can be described as

$$F = \tau_w A_c = K\dot{\gamma}_w^n A_c, \tag{4.4}$$

where $\tau_w$ and $\dot{\gamma}_w$ are the shear stress and shear rate at the capillary wall; $K$ and $n$ are the particulate suspension consistency and behaviour indices (respectively); and $A_c$ is the capillary surface area in contact with the suspension. The total shear force is the result of the contributions from the individual suspension components. Specifically, particles near the capillary wall influence the fluid flow near the wall by diverting flow between the wall and the particles as illustrated in figure 8.

The total flow is the combination of the effects from regions ($A_i$) where particles are influencing the flow near the wall and the rest of the capillary wall region where the fluid flow effects dominate. Hence, the total force on the capillary wall can also be obtained as

$$F = \sum F_{im} + F_{un}, \tag{4.5}$$

where $F_{im}$ are the wall forces generated under the influence of particles near the wall and $F_{un}$ are the forces generated without particle influence.

To describe the characteristics of wall flow with and without particle influence, the distance between a particle and the centreline of the capillary is defined as $q$. The radius of the inner boundary of the region where the particles may influence the wall shear rate is defined as $q_0$. For any particle where $q < q_0$, the wall shear rate will not be influenced by the particle and the shear rate is defined as $\dot{\gamma}_{w,0}$. For a particle where $q > q_0$, the wall shear rate will be influenced by the particle, and the particle is defined as a wall particle. The average wall shear rate influenced by the $i$th wall particle is defined as $\overline{\dot{\gamma}}_{w,i}$, and the area on the wall where the shear rate is influenced is defined as $A_i$, as illustrated in figure 8.

Based on the above description of the flow, the force on the capillary wall can be derived as

$$F = \sum_{i=1}^{N_w} K_f (\overline{\dot{\gamma}}_{w,i})^{n_f} A_i + K_f \dot{\gamma}_{w,0}^{n_f} \left( 1 - \sum_{i=1}^{N_w} A_i \right), \tag{4.6}$$

where $K_f$ and $n_f$ are the flow consistency and behaviour indices (respectively) for the fluid, and $N_w$ is the number of particles near the wall ($q > q_0$). Since the total force on the capillary wall is the same in equations (4.4) and (4.6), the relationship between $K$ and $K_f$ can be derived as

$$\frac{K}{K_f} = \left( 1 + \sum_{i=1}^{N_w} \left( \left( \frac{\overline{\dot{\gamma}}_{w,i}}{\dot{\gamma}_{w,0}} \right)^{n_f} - 1 \right) \frac{A_i}{A_c} \right) \frac{\dot{\gamma}_{w,0}^{n_f}}{\dot{\gamma}_w^n}. \tag{4.7}$$

Hence, $K/K_f$ is determined by the four terms: $\dot{\gamma}_{w,0}^{n_f}/\dot{\gamma}_w^n$, $\overline{\dot{\gamma}}_{w,i}/\dot{\gamma}_{w,0}$, $A_i/A_c$ and $N_w$. To analyse the relation between $K/K_f$ and properties of the particles, properties of the suspension fluid, and $D$, these four terms will be discussed separately below.

*Analysis for $\dot{\gamma}_{w,0}^{n_f}/\dot{\gamma}_w^n$:* as noted in §b, the fully developed suspension can be regarded as uniform at the free-flow condition, so the velocity profile is expected to have a universal shape for different $D$. Hence, $\dot{\gamma}_{w,0}$ and $\dot{\gamma}_w$ are also constants at the free-flow condition after non-dimensionalizing. Since $n$ with the same suspension fluid is a function of $\phi$ only as discussed above, it can be concluded that $\dot{\gamma}_{w,0}^{n_f}/\dot{\gamma}_w^n$ is a function only of $\phi$, denoted as $G(\phi)$. Using equation (4.1), $G(\phi)$ can be represented as

$$G(\phi) = \frac{\dot{\gamma}_{w,0}^{n_f}}{\dot{\gamma}_w^n} = \frac{\dot{\gamma}_{w,0}^{n_f}}{\dot{\gamma}_w^{n_f(1+a\phi)}} = \frac{\dot{\gamma}_{w,0}^{n_f}}{\dot{\gamma}_w^{n_f}} \dot{\gamma}_w^{-n_f a\phi} = \frac{\dot{\gamma}_{w,0}^{n_f}}{\dot{\gamma}_w^{n_f}} e^{\ln(\dot{\gamma}_w^{-n_f a\phi})} = \frac{\dot{\gamma}_{w,0}^{n_f}}{\dot{\gamma}_w^{n_f}} e^{-n_f a\phi \ln(\dot{\gamma}_w)} = C_1 e^{b_1 \phi}, \tag{4.8}$$

where $C_1$ and $b_1 = -n_f a \ln(\dot{\gamma}_w)$ are constants. Combining equations (4.7) and (4.8) gives

$$\frac{K}{K_f} = G(\phi) f_s, \tag{4.9}$$

with

$$f_s = 1 + \sum_{i=1}^{N_w} \left( \left( \frac{\overline{\dot{\gamma}}_{w,i}}{\dot{\gamma}_{w,0}} \right)^{n_f} - 1 \right) \frac{A_i}{A_c}. \tag{4.10}$$

*Analysis for $\overline{\dot{\gamma}}_{w,i}/\dot{\gamma}_{w,0}$:* the term $\overline{\dot{\gamma}}_{w,i}/\dot{\gamma}_{w,0}$ is the ratio of wall shear rates for the cases with and without particles near the capillary wall. As the fluid in the capillary is the same with only the radial position of the particles relative to the wall changing, the boundary flow profile shape near the capillary wall can be assumed 'similar' for the two cases. That is, the velocity profile near the wall is modelled as

$$u = UH \left( \frac{y}{y_m} \right), \tag{4.11}$$

for $0 < y < y_m$ where $y = R - q$ and $y_m$ is the radial distance from the capillary wall at which the flow reaches the centerline velocity $U$, and $H$ is a function describing the velocity profile shape. Then the shear rate of the boundary layer can be expressed as

$$\dot{\gamma} = \frac{\partial u}{\partial y} = H'(0) \frac{U}{y_m}. \tag{4.12}$$

Assuming the particle influence on the shear rate scales with particle size gives

$$(y_m)_0 \equiv R - q_0 = \epsilon_0 r \tag{4.13}$$

and

$$(y_m)_i \equiv R - q_i = \epsilon_i r, \tag{4.14}$$

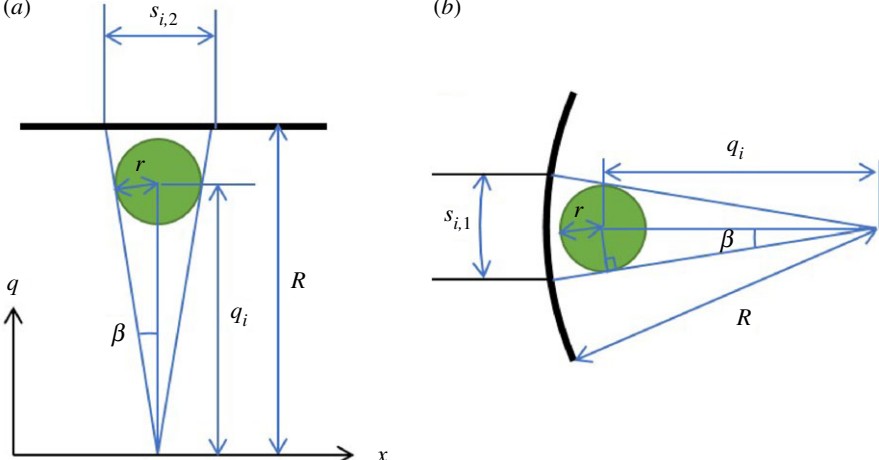

**Figure 9.** Schematic diagram of the $i$th wall particle and its geometrical relation with the capillary and influenced area in the ($a$) axial direction and ($b$) radial direction. (Online version in colour.)

where $r$ is the particle radius, and $\epsilon_0$ and $\epsilon_i$ are determined by the flow conditions. Then using equation (4.12), the term $\overline{\dot\gamma}_{w,i}/\dot\gamma_{w,0}$ can be expressed as

$$\frac{\overline{\dot\gamma}_{w,i}}{\dot\gamma_{w,0}} = \frac{\epsilon_0}{\epsilon_i}. \tag{4.15}$$

*Model for $A_i/A_c$:* for the region of particle influence, the $i$th wall particle can affect wall shear within the area of $A_i$ on the wall as illustrated in figure 9.

Based on the geometrical relations shown in figure 9 and the approximation $\beta \approx \tan \beta = r/q_i$ for $D/d > 20$, the following geometrical relations and approximations can be found.

$$\frac{A_i}{A_c} = \frac{(\pi/4)s_{i,1}s_{i,2}}{2\pi RL} = \frac{(\pi/4)(2\beta R)(2R\tan\beta)}{2\pi RL} = \frac{R}{2L}\left(\frac{r}{q_i}\right)^2 = \frac{R}{2L}\left(\frac{r}{R-\epsilon_i r}\right)^2. \tag{4.16}$$

This construction emphasizes that the dominant influence of the particle narrows to the region just between the particle and the wall as particles approach the wall, where the effect on the wall shear stress is greatest.

*Model for $N_w$:* as the value of $\epsilon_i$ can vary with particle location within the capillary and the total number of wall particles ($N_w$) is large, using $\bar\epsilon$ as the average of $\epsilon_i$ in the above results gives

$$f_s = 1 + \sum_{i=1}^{N_w}\left(\left(\frac{\epsilon_0}{\epsilon_i}\right)^{n_f} - 1\right)\frac{R}{2L}\left(\frac{r}{R-\epsilon_i r}\right)^2 = 1 + N_w\left(\left(\frac{\epsilon_0}{\bar\epsilon}\right)^{n_f} - 1\right)\frac{R}{2L}\left(\frac{r}{R-\bar\epsilon r}\right)^2. \tag{4.17}$$

Based on the geometry of the capillary and particles, the total number of particles in the capillary, $N_p$, the total number of wall particles can be expressed as

$$N_w = N_p P_w = \frac{3}{4}\phi L\frac{R^2}{r^3} = \frac{3\phi L}{4R}\omega^3 P_w, \tag{4.18}$$

where $P_w$ is the probability that a particle is in the region influencing the wall shear rate ($q > q_0$).

As the particles are uniformly distributed inside the capillary on average, $P_w$ can be calculated based on geometrical considerations. Calculation of $P_w$ in the actual three-dimensional case can be simplified to the calculation of the probability for an equivalent two-dimensional projection onto the capillary cross section as illustrated in figure 10$a$. The probability that a particle is in the

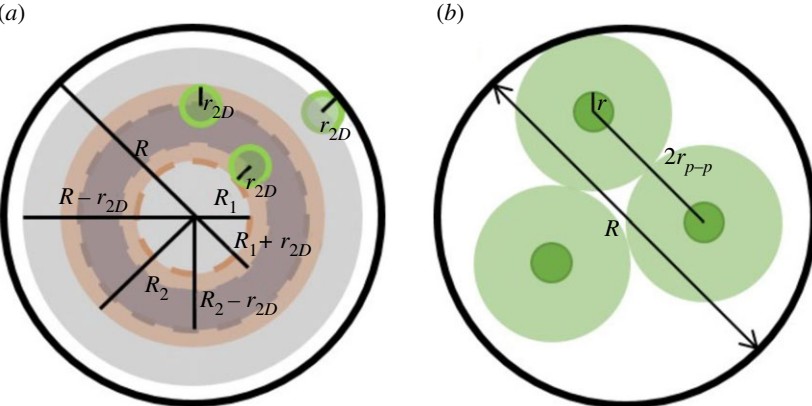

**Figure 10.** (*a*) Schematic diagram of particles randomly distributed in a ring area and dimensions related to different regions that may contain particles. (*b*) Schematic diagram of particles and their interaction spheres under dense packing. (Online version in colour.)

ring area between the radii of $R_1$ and $R_2$ can be determined as the ratio of the total accessible area in the ring to that of the capillary, namely,

$$P_{R_1 \leftrightarrow R_2} = \frac{(R_2 - r_{2D})^2 - (R_1 + r_{2D})^2}{(R - r_{2D})^2}, \tag{4.19}$$

where $R$ is the radius of the capillary, and $r_{2D}$ is the adjusted particle radius for the two-dimensional case. (For behaviour with a non-uniform particle distribution, equation (4.19) can be modified accordingly.) For the two-dimensional case, the particles are treated as cylinders of length $2r$, so the equivalent radius is determined as $r_{2D} = \sqrt{2/3}r$ by matching particle volume, namely, $(2r)\pi r_{2D}^2 = \frac{4}{3}\pi r^3$. Hence, $N_w$ can be derived as

$$N_w = N_p P_w = \frac{3\phi L}{4R} \omega^3 \frac{(2\omega - \epsilon_0)\left(\epsilon_0 - 2\sqrt{2/3}\right)}{\left(\omega - \sqrt{2/3}\right)}. \tag{4.20}$$

Using the above results, $f_s$ for the free-flow condition ($\omega > \omega_{pic}$) can be modelled as

$$f_s = 1 + \frac{3\phi}{8}\left(\epsilon_0 - 2\sqrt{\frac{2}{3}}\right)\left(\left(\frac{\epsilon_0}{\bar{\epsilon}}\right)^{n_f} - 1\right)\frac{\omega^3(2\omega - \epsilon_0)}{(\omega - \bar{\epsilon})^2\left(\omega - \sqrt{\frac{2}{3}}\right)^2}, \tag{4.21}$$

where $\epsilon_0$, $\bar{\epsilon}$ and $\omega_{pic}$ are determined by the suspension properties.

*Determination of $\omega_{pic}$, $\epsilon_0$ and $\bar{\epsilon}$*: particles begin interfering with each other to increase $K$ under the particle interaction condition ($\omega < \omega_{pic}$). The interaction comes from the forces generated between adjacent particles by the flow between them when the separation distance is small enough, resulting in rapid increasing of the flow consistency index. As the interaction is perpendicular to the flow direction, the particle separation distance causing the interaction is in the radial direction as illustrated in figure 10*b*. The fluid within the region of particle interaction is represented by a sphere with the radius of $r_{p-p}$. When the distance between two particles is smaller than $2r_{p-p}$, they are considered to interact.

Based on these assumptions, when $\omega = \omega_{pic}$ (where particle interaction starts), the relationship between $r_{p-p}$ and the capillary dimensions is given by

$$N_{2D}\pi r_{p-p}^2 = \phi_{SP}\pi R^2, \tag{4.22}$$

where $\phi_{SP}$ is the sphere volume ratio for close packing in the capillary (cylinder) and $N_{2D}$ is the number of particles in the capillary cross section. In this case ($\omega = \omega_{pic}$), $\omega$ is large enough that the boundary spheres around the particles can be regarded as densely packed inside the capillary.

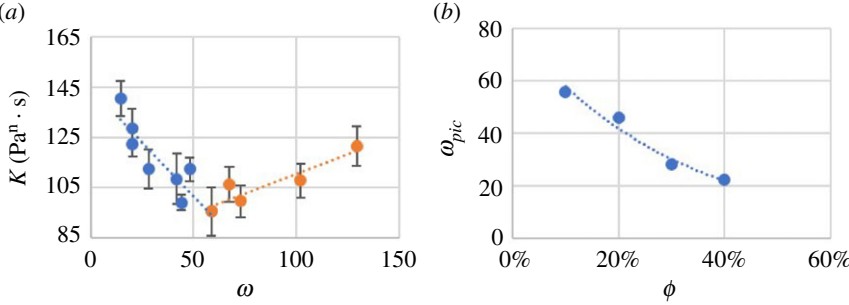

**Figure 11.** (a) Example of $\omega_{pic}$ obtained from $K(\text{Pa}^n \cdot s)$ and $\phi$ trendlines (when $\phi = 10\%$); (b) $\omega_{pic}$ versus $\phi$ for experimental results for UV-225-1-based suspensions. (Online version in colour.)

According to the geometrical relationship between the particle and capillary cross sections, $N_{2D}$ can be derived as $N_{2D} = \phi \pi R^2 / \pi r_{2D}^2 = 3\phi R^2 / 2r^2 = (3\phi/2)\omega^2$. As the particles are constrained in the capillary with the radius of $R$, it is reasonable to assume when $\alpha_{p-p} \equiv R/r_{p-p}$ arrives at a characteristic value, $\alpha_{pic}$, the particle interaction condition is achieved. Hence, the relation between $\omega_{pic}$ and the particle volume fraction $\phi$ has the following form:

$$\omega_{pic} = \frac{C_2}{\sqrt{\phi}}, \quad \text{where } C_2 = \alpha_{pic}\sqrt{\frac{2}{3}\phi_{SP}}. \tag{4.23}$$

As the options of capillary and particle diameter were limited, $\omega_{pic}$ could not be directly detected from the experiments. Instead, $\omega_{pic}$ was approximated by the intersection of linear best-fit lines on both sides of the minimum in $K$. An example for $\phi = 10\%$ is illustrated in figure 11a. A least-square fit of equation (4.23) to the resulting values for $\omega_{pic}$ in suspensions using UV 225-1 gives $C_2 = 18.5 \pm 0.2$. The uncertainties were calculated from the Jackknife method [43]. The results of this fit are shown in figure 11b.

The $\epsilon_0 r$ term is the largest distance from the capillary wall for which particles may affect the wall shear rate. At the end of the free-flow condition ($\omega = \omega_{pic}$), the flow has the largest number of particles impacting the wall shear rate. To simplify the calculation, it can be assumed that all the particles affect the wall shear rate at this point. Hence, the governing radius of the sphere is the same as the capillary radius. That is, at $\omega = \omega_{pic}$, $\epsilon_0 r = R$. Hence,

$$\epsilon_0 = \omega_{pic} = C_2\phi^{-\frac{1}{2}} \tag{4.24}$$

The average distance between particles and the capillary wall is given by $\bar{\epsilon}r$. For real flows, many factors may impact on this value. To accommodate this complexity, $\bar{\epsilon}$ is determined empirically using measurements of $K_{min}/K_{inf}$ where $K_{min}$ is the minimum value of $K$ when $\omega = \omega_{pic}$ for a given $\phi$ and $K_{inf}$ is the value of $K$ when $\omega \to \infty$, which can be regarded as a constant for the same $\phi$.

Values of $K_{inf}$ at $\phi = 30\%$ and $40\%$ were directly measured from data in figure 4a. However, $K_{inf}$ could not be measured directly at $\phi = 10\%$ and $20\%$ since the data did not reach a constant value in the range of $\omega$ tested. Instead, values of $K_{inf}$ at $\phi = 10\%$ and $20\%$ were extrapolated from a second-order polynomial model (an exponential model provided similar results). Determination of $K_{min}$ was done simultaneously with $\omega_{pic}$ following the method illustrated in figure 11a.

$K_{min}/K_{inf}$ can be expressed as a function of $\bar{\epsilon}$ and $\phi$ using equation (4.21). A least-square fit of this result for the determined values of $K_{min}$ and $K_{inf}$ at each $\phi$ was used to find $\bar{\epsilon}$. As $\bar{\epsilon}$ is closely related with $\epsilon_0$, an empirical power law relationship is proposed for this parameter

$$\bar{\epsilon} = C_3\phi^{b_3}, \tag{4.25}$$

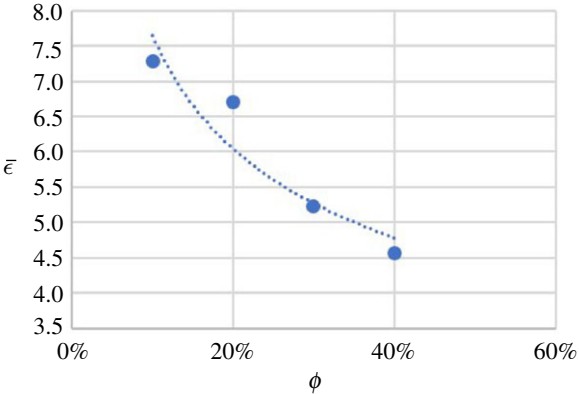

**Figure 12.** $\bar{\epsilon}$ versus $\phi$ (UV 225-1-based suspensions). (Online version in colour.)

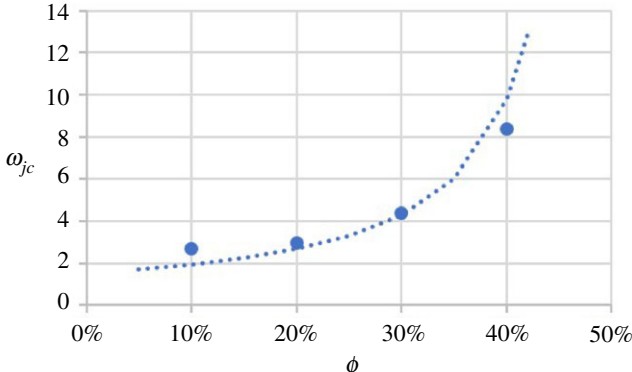

**Figure 13.** $\omega_{jc}$ versus $\phi$ (experimental results for UV 225-1-based suspensions). (Online version in colour.)

where $C_3$ and $b_3$ are two adjustable constants. A least-square fit of equation (4.25) to the UV 225-1-based suspension gives $C_3 = 3.5 \pm 0.4$ and $b_3 = -0.34 \pm 0.11$ as shown in figure 12.

Using the model developed above, $f_s$ at $\omega = \infty$ (constant $K$) can be calculated as 20.3, 24.1, 26.1 and 27.2 for $\phi = 10\%, 20\%, 30\%$ and $40\%$, respectively. With measured $K$ and $K_f$, the values of $G$ were obtained as 6.6, 7.2, 9.1 and 10.7 for $\phi = 10\%, 20\%, 30\%$ and $40\%$, respectively. A least-square fit of equation (4.8) gives $C_1 = 5.6 \pm 0.4$ and $b_1 = 1.6 \pm 0.2$.

Combining the above results, the complete model for $K$ in the free-flow condition is given by

$$\frac{K}{K_f} = C_1\, e^{b_1 \phi} \left[ 1 + \frac{3\phi}{8} \left( \left( \frac{C_2}{C_3} \phi^{(1/2)-b_3} \right)^{n_f} - 1 \right) \left( C_2 \phi^{1/2} - 2\sqrt{\frac{2}{3}} \right) \frac{\omega^3 (2\omega - C_2 \phi^{1/2})}{(\omega - C_3 \phi^{b_3})^2 \, (\omega - \sqrt{2/3})} \right].$$

(4.26)

A plot of this model with the experimental data is included in figure 14, which shows that the model generally follows the trends in the data well for $\omega > \omega_{pic}$. The possible reasons for differences between the model and experiments (aside from experimental uncertainty and uncertainty in the model-fitted parameters) are likely related to simplifications used to obtain the model. Ignored relative positions among particles, such as particles located adjacent to or sheltered by others, and their polydisperse character, may lead to complex packing density and other effects and result in higher or lower flow consistency index on the wall.

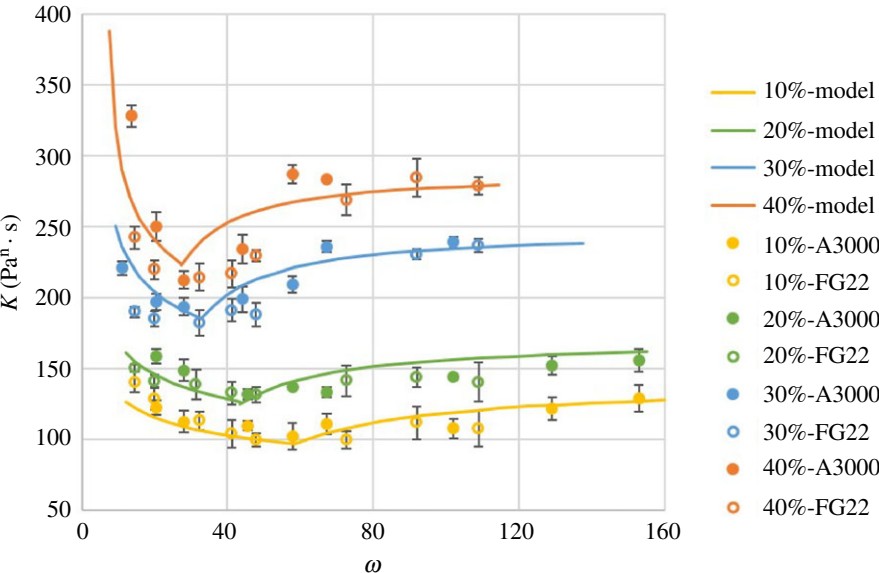

**Figure 14.** Flow consistency index $K$ (Pa$^n$ · s) versus $\omega$ (experimental results for UV 225-1-based suspensions) under both flow conditions with legend indications following the format: [particle volume fraction] − [particle type]. (Online version in colour.)

### (ii) Model for the particle interaction condition

As shown in figure 4$a$, $K$ increases with decreasing $\omega$ for the particle interaction condition ($\omega_{jc} < \omega < \omega_{pic}$). This behaviour results from the interaction of nearby particles as flow space becomes restricted. The interaction becomes stronger with smaller distance between particles, which is controlled by $\omega$ and $\phi$. Given the likely complex nature of this interaction that is difficult to visualize, the behaviour in the particle interaction condition is modelled using an empirical relationship as

$$K = \frac{A}{(\omega - \omega_{jc})^b}, \tag{4.27}$$

where $\omega_{jc}$ is the value of $\omega$ when jamming occurs ($K \to \infty$), and $A$ and $b$ are curve-fitting parameters. As the equation is valid within $\omega_{jc} < \omega < \omega_{pic}$, $A$ can be derived as $A = K_{min}(\omega_{pic} - \omega_{jc})^b$ in order to match the model for the free-flow condition at $\omega_{pic}$. Hence, equation (4.27) can be expressed as

$$\frac{K}{K_{min}} = \left( \frac{\omega_{pic} - \omega_{jc}}{\omega - \omega_{jc}} \right)^b. \tag{4.28}$$

*Model for $\omega_{jc}$ and $b$*: the value of $\omega_{jc}$ is difficult to determine directly, which was compounded by limited options for capillary and particle diameters. So instead, it was determined, together with $b$, by fitting equation (4.28) to the data for $\omega < \omega_{pic}$. A least-square fit of equation (4.28) was used to determine $b$ and $\omega_{jc}$ for each $\phi$. The values for $b$ were similar in all cases, so $b$ is taken as the average value of 0.21. The values for $\omega_{jc}$ were obtained as 2.7, 3.0, 4.4 and 8.4 for $\phi = 10\%, 20\%, 30\%$ and $40\%$, respectively, and are plotted in figure 13.

According to the experimental results, for constant $\phi$ jamming occurs at larger $\omega$ as $\phi$ is increased. This behaviour is expected from behaviour at limiting conditions. For the limiting case of only one particle in the suspension, $\phi = 0$. However, jamming can still happen with only one particle in the capillary if the particle is the same diameter as the capillary. That is, $\omega_{jc} \to 1$ as $\phi \to 0$. With more particles added into the suspension, $\phi$ increases and will approach its maximum particle volume fraction, $\phi_M$, for which each particle is in contact with multiple other particles and the liquid matrix fills the voids. At this condition, the particle configuration

is stable even under the action of finite loads and the suspension has become jammed even with a very large capillary. For this condition, $\omega_{jc} \to \infty$. The value of $\phi_M$ may be impacted by many factors, including the flow state and polydispersity [25,26,44]. To simplify the analysis, it will be assumed that $\phi_M \approx 0.585$, which is appropriate for monodisperse spheres [27] and is close to the range of values for bidisperse spheres [26]. To capture the dependence of $\omega_{jc}$ on $\phi$ (for a fixed $\phi$), it is represented as

$$\omega_{jc} = \frac{C_4}{(\phi_M - \phi)^{b_4}},\tag{4.29}$$

where $C_4$ and $b_4$ are two adjustable parameters. For UV 225-1-based suspensions, a least-square fit of equation (4.29) to the results for $\omega_{jc}$ gives the values of $C_4 = 0.55 \pm 0.16$ and $b_4 = 1.62 \pm 0.28$ with the resulting curve fit shown in figure 13.

Hence, from equations (4.28) and (4.29), $K$ under the particle interaction condition is modelled as

$$\frac{K}{K_{min}} = \left( \frac{C_2 \phi^{-1/2} - C_4(\phi_M - \phi)^{-b_4}}{\omega - C_4(\phi_M - \phi)^{-b_4}} \right)^b,\tag{4.30}$$

where $K_{min}$ is calculated from equations (4.24) and (4.26). This equation is plotted in figure 14 for $\omega < \omega_{pic}$ and follows the measured results well for this region.

*Particle diameter distribution impact and jamming*: besides $\phi$ and $\omega$, there is another possible factor influencing the suspension rheology under the particle interaction condition. As presented in figure 4a, the $K$ values may differ when $\omega$ is the same but the particles are different. This may be caused by different particle diameter distributions. It is also noted that suspensions of the same particle volume fraction with particles having a wider particle diameter distribution (A3000 particles) have a larger $K$ for the same $\omega$, especially for $\omega < \omega_{pic}$. A possible explanation is that the larger particles dominate the interaction. It is also found that this trend is more significant with a higher particle volume fraction, as the interaction is stronger when there are more particles to interact.

Jamming occurs when particles begin to cluster and span the entire capillary. The most important parameter describing jamming is $\omega_{jc}$. An empirical model proposed in equation (4.29) captures the basic behaviour. It shows $\omega_{jc}$ increasing with $\phi$ increasing as expected, meaning jamming tends to occur with a larger difference between the capillary diameter and the mean particle diameter when the particle volume fraction is higher.

One key characteristic of jamming is that it does not always happen at $\omega_{jc}$. As $\omega_{jc}$ is derived from the semi-empirical process, the impact of the particle diameter distribution is ignored. In the real situation, the particles at any given location within the flow have different diameters. The particle clustering that spans the capillary cross section and halts the flow is dominated by the larger particles. Hence, jamming has a higher probability of happening when the particle diameter range is larger for the same $\omega$ and $\phi$. For example, when $\omega$ was reduced, jamming happened in three out of three tests at similar $\omega$ with the suspension made of UV 225-1 and $\phi = 30\%$ using the FG22 particles and happened in two out of three tests with the suspension made of UV 225-1 and $\phi = 30\%$ using the A3000 particles.

Finally, the impact of particle diameter distribution on jamming is stronger with larger $\phi$. The possibility of particle jamming was different for the $\phi = 30\%$ suspensions as discussed above. However, there was almost no difference in jamming with FG22 and A3000 at $\phi = 10\%$. This behaviour is also expected following the trends observed for the particle interaction condition, which has a much weaker interaction for smaller $\phi$.

## 5. Conclusion

In this work, the rheology of particulate suspensions in shear-thinning fluids was investigated, concentrating on the flow behaviour ($n$) and consistency ($K$) indices. Related theories and models on viscosity were reviewed and compared. Experiments to investigate the indices in a confined environment similar to extrusion through a small nozzle were designed and conducted. Particle

volume fraction ($\phi$) and the ratio of capillary inside diameter to the particle mean diameter ($\omega$) were found to be the key factors impacting the suspension rheology.

Based on the experimental results, the flow behaviour index $n$ was found to be only dependent on $\phi$ for a suspension with the same suspension fluid. A linear relation between $n$ and $\phi$ was observed and modelled with an empirical equation with acceptable accuracy. The primary difference observed between suspensions with Newtonian and shear-thinning suspension fluids was that adding particles to a shear-thinning suspension fluid made the suspension more shear thinning while a suspension in a Newtonian suspension fluid remained Newtonian.

Experimental results showed that the behaviour of $K$ for suspensions with different $\phi$ followed similar trends with $\omega$. The behaviour was classified into two categories: the free-flow condition and the particle interaction condition. Analysis of the flow behaviour attributing differing effects based on particle proximity with the wall in the free-flow condition produced a model for this behaviour with few empirical parameters and a corresponding empirical model for the particle interaction condition was constructed. With all the models together, the flow consistency index of the particulate suspension with different $\phi$ and $\omega$ can be described with acceptable accuracy, as illustrated in figure 14. Differences between the model and tested results were also discussed in terms of the approximations made in the model. In particular, it was noted that flow with the same $\phi$ and $\omega$ had a larger $K$ when the particles had a wider diameter distribution. Such particle diameter distribution effects likely had an effect on jamming as well, with some limited data indicating a higher $\phi$ and a wider particle diameter distribution may result in a higher probability of jamming occurring.

Using the obtained models in AM applications, the manufacturing process can be improved. Based on the requirements of accuracy and the particle volume fraction of the particulate composites, the extruder nozzle ID and particle mean diameter can be properly selected and optimized. Jamming can be avoided by keeping the ratio of extruder nozzle ID to the particle mean diameter larger than the jamming ratio, $\omega_{jc}$, for the chosen particle volume fraction. Relatively small extruding force can be achieved by keeping $\omega$ close to the ratio of the intersection of two flow conditions, $\omega_{pic}$. Minimizing repeated work caused by jamming and unsatisfactory material properties can improve manufacturing efficiency and quality.

However, additional work is necessary to apply the models to AM situations. Temperature effects were avoided in this investigation by using silicone, and the rheology of polymer melts used in AM is strongly dependent on temperature [3,45]. Also, the extruder nozzle is shorter with a more complex geometry compared with the capillaries used in this investigation. Additional work considering appropriate temperature and geometry corrections would be necessary to accurately apply these models to the range of conditions encountered in AM, but this work provides the framework and establishes the key factors for consideration in these applications.

**Data accessibility.** The processed experimental data are available on the SMU research archive, SMU Scholar, at https://scholar.smu.edu/engineering_mechanical_research/6/.

**Authors' contributions.** B.X.: conceptualization, data curation, formal analysis, investigation, methodology, software, validation, visualization, writing—original draft; P.K.: conceptualization, funding acquisition, investigation, methodology, validation, writing—review and editing.

Both authors gave final approval for publication and agreed to be held accountable for the work performed therein.

**Conflict of interest declaration.** We declare we have no competing interests.

**Funding.** This paper is based on the work supported by the National Science Foundation under grant no. 1317961. The first author, B.X., was supported by a Moody Dissertation Fellowship from the SMU Moody Graduate School during the last year of this investigation.

**Acknowledgements.** Research was performed at the Department of Mechanical Engineering, Southern Methodist University, Dallas, Texas. The authors also appreciate for their assistance: Prof. Xu Nie, Dr Qiran Sun, Dr Matt Saari, Travis Mayberry and Gorkem Guclu.

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
