## [Peer Review File · Proceedings. Mathematical, Physical, and Engineering Sciences]

Review History

RSPA-2021-0615.R0 (Original submission)

Review form: Referee 1

Is the manuscript an original and important contribution to its field?

Good

Is the paper of sufficient general interest?

Acceptable

Is the overall quality of the paper suitable?

Acceptable

Can the paper be shortened without overall detriment to the main message?

Yes

Do you think some of the material would be more appropriate as an electronic appendix?

No

Do you have any ethical concerns with this paper?

No

Recommendation?

Accept with minor revision (please list in comments)

Comments to the Author(s)

Rheology of Particulate Suspensions with Non-Newtonian Fluids in Capillaries – Xia and Krueger

The paper reports experiments on particulate-shear thinning fluid suspensions in capillaries, as a 'model' of additive manufacturing (3D printing) applications.

Comments

Line 52 – mention that much previous work looks at shear thickening 'material' – however it is important here to relate that thickening is usually the result of the particulate phase, rather than a particulate phase in a thickening medium. Given the potentially separate roles of medium rheology and the particulate phase care should be taken to be clear on what is thickening, what is thinning...

On this point, the choice of literature referred to regarding the 'jamming' role of the particle phase is a bit eclectic – there are lots of more recent works by Wyart, Cates, etc representing probably the most recent work on high concentration particle suspension rheology (albeit generally in Newtonian fluids) and the relevance of this work ought to be considered in the paper.

Was there any evidence of issues with the contraction from syringe into capillary especially at high particle fraction? Previous work (eg Isa, Haw etc) has shown significant impacts of contraction and greater degree of 'jamming' due to the extensional flow. It is somewhat surprising this is not observed at 40% fraction for example and it would be good to have a clear confirmation that this is the case. One of the issues with such jamming is that the fluid filters through the jammed particles (eg Haw 2004) and so the downstream volume fraction is decreased.

Overall there is not much clear conclusion about the specific role of non-Newtonian medium – how does it create differences compared to Newtonian medium? This seems to be the point of the paper at the start but it is not clearly concluded on. This ought to be improved to help readers grasp the key outcome even if they are not going to follow the details of the models presented.

Review form: Referee 2

Is the manuscript an original and important contribution to its field?

Good

Is the paper of sufficient general interest?

Good

Is the overall quality of the paper suitable?

Excellent

Can the paper be shortened without overall detriment to the main message?

Yes

Do you think some of the material would be more appropriate as an electronic appendix?

No

Do you have any ethical concerns with this paper?

No

Recommendation?

Major revision is needed (please make suggestions in comments)

Comments to the Author(s)

I would suggest the authors to go over my attached comments. Please see Appendix A.

Review form: Referee 3**Is the manuscript an original and important contribution to its field?**

Acceptable

Is the paper of sufficient general interest?

Marginal

Is the overall quality of the paper suitable?

Poor

Can the paper be shortened without overall detriment to the main message?

Yes

Do you think some of the material would be more appropriate as an electronic appendix?

No

Do you have any ethical concerns with this paper?

No

Recommendation?

Reject – article is scientifically unsound

Comments to the Author(s)

The authors investigate the effective viscosity of particle suspensions in non-Newtonian fluids moving inside capillaries. The authors present a thorough experimental investigation that would find an interested audience in a more technical journal. The authors then develop a mesoscopic model to describe the observed results. The model has several problematic assumptions, as detailed below.

1. The authors separate free-flow and particle-interaction regions depending on the particle-to-capillary size ratio ' w '. This is probably incorrect. At high concentrations there should be particle-particle interactions independent of the value of ' w '.
2. The authors claim that experiments have shown that the distribution of particles is uniform in the cross section for large ' w ' values. This is incorrect. There is a large body of work on suspension flows, including several theoretical models, describing the non-uniform distribution of particles in the cross section.

Other issues:

- 3- The authors discuss jamming as dependent on ' w '. In general, it has to depend on the concentration as well. For low enough concentrations, even ' w ' close to 1 would not lead to

jamming.

4- The authors calculate the length to reach fully developed flow ignoring particle migration. However, the concentration is not uniform, as stated above, and the length it takes to become fully developed needs to be considered.

5- The authors estimate an area of influence of the particles on the wall considering a geometrical projection that completely ignores the hydrodynamics of the problem. The area on which a particle affects the flow could be significantly larger.

Decision letter (RSPA-2021-0615.R0)

06-Jan-2022

Dear Dr Xia

The Editor of Proceedings A has now received comments from referees on the above paper and would like you to revise it in accordance with their suggestions which can be found below (not including confidential reports to the Editor).

Please submit a copy of your revised paper within four weeks - if we do not hear from you within this time then it will be assumed that the paper has been withdrawn. In exceptional circumstances, extensions may be possible if agreed with the Editorial Office in advance.

Please note that it is the editorial policy of Proceedings A to offer authors one round of revision in which to address changes requested by referees. If the revisions are not considered satisfactory by the Editor, then the paper will be rejected, and not considered further for publication by the journal. In the event that the author chooses not to address a referee's comments, and no scientific justification is included in their cover letter for this omission, it is at the discretion of the Editor whether to continue considering the manuscript.

To revise your manuscript, log into <http://mc.manuscriptcentral.com/prsa> and enter your Author Centre, where you will find your manuscript title listed under "Manuscripts with Decisions." Under "Actions," click on "Create a Revision." Your manuscript number has been appended to denote a revision.

You will be unable to make your revisions on the originally submitted version of the manuscript. Instead, revise your manuscript and upload a new version through your Author Centre.

When submitting your revised manuscript, you will be able to respond to the comments made by the referee(s) and upload a file "Response to Referees" in Step 1: "View and Respond to Decision Letter". Please provide a point-by-point response to the comments raised by the reviewers and the editor(s). A thorough response to these points will help us to assess your revision quickly. You can also upload a 'tracked changes' version either as part of the 'Response to reviews' or as a 'Main document'.

IMPORTANT: Your original files are available to you when you upload your revised manuscript. Please delete any unnecessary previous files before uploading your revised version.

When revising your paper please ensure that it remains under 28 pages long. In addition, any pages over 20 will be subject to a charge (£150 + VAT (where applicable) per page). Your paper has been ESTIMATED to be 19 pages.

Open Access

You are invited to opt for open access, our author pays publishing model. Payment of open access fees will enable your article to be made freely available via the Royal Society website as soon as it is ready for publication. For more information about open access please visit <https://royalsociety.org/journals/authors/open-access/>. The open access fee for this journal is £1700/\$2380/€2040 per article. VAT will be charged where applicable. Please note that if the corresponding author is at an institution that is part of a Read and Publishing deal you are required to select this option. See <https://royalsociety.org/journals/librarians/purchasing/read-and-publish/read-publish-agreements/> for further details.

Once again, thank you for submitting your manuscript to Proc. R. Soc. A and I look forward to receiving your revision. If you have any questions at all, please do not hesitate to get in touch.

Yours sincerely
Raminder Shergill
proceedingsa@royalsociety.org

on behalf of
Dr Paul Grassia
Board Member
Proceedings A

Reviewer(s)' Comments to Author:

Referee: 1

Comments to the Author(s)

Rheology of Particulate Suspensions with Non-Newtonian Fluids in Capillaries – Xia and Krueger

The paper reports experiments on particulate-shear thinning fluid suspensions in capillaries, as a 'model' of additive manufacturing (3D printing) applications.

Comments

Line 52 – mention that much previous work looks at shear thickening 'material' – however it is important here to relate that thickening is usually the result of the particulate phase, rather than a particulate phase in a thickening medium. Given the potentially separate roles of medium rheology and the particulate phase care should be taken to be clear on what is thickening, what is thinning...

On this point, the choice of literature referred to regarding the 'jamming' role of the particle phase is a bit eclectic – there are lots of more recent works by Wyart, Cates, etc representing probably the most recent work on high concentration particle suspension rheology (albeit generally in Newtonian fluids) and the relevance of this work ought to be considered in the paper. Was there any evidence of issues with the contraction from syringe into capillary especially at high particle fraction? Previous work (eg Isa, Haw etc) has shown significant impacts of contraction and greater degree of 'jamming' due to the extensional flow. It is somewhat surprising this is not observed at 40% fraction for example and it would be good to have a clear confirmation that this is the case. One of the issues with such jamming is that the fluid filters through the jammed particles (eg Haw 2004) and so the downstream volume fraction is decreased.

Overall there is not much clear conclusion about the specific role of non-Newtonian medium – how does it create differences compared to Newtonian medium? This seems to be the point of the paper at the start but it is not clearly concluded on. This ought to be improved to help readers grasp the key outcome even if they are not going to follow the details of the models presented.

Referee: 2

Comments to the Author(s)

I would suggest the authors to go over my attached comments. Please see Appendix A.

Referee: 3

Comments to the Author(s)

The authors investigate the effective viscosity of particle suspensions in non-Newtonian fluids moving inside capillaries. The authors present a thorough experimental investigation that would find an interested audience in a more technical journal. The authors then develop a mesoscopic model to describe the observed results. The model has several problematic assumptions, as detailed below.

1. The authors separate free-flow and particle-interaction regions depending on the particle-to-capillary size ratio ' w '. This is probably incorrect. At high concentrations there should be particle-particle interactions independent of the value of ' w '.
2. The authors claim that experiments have shown that the distribution of particles is uniform in the cross section for large ' w ' values. This is incorrect. There is a large body of work on suspension flows, including several theoretical models, describing the non-uniform distribution of particles in the cross section.

Other issues:

- 3- The authors discuss jamming as dependent on ' w '. In general, it has to depend on the concentration as well. For low enough concentrations, even ' w ' close to 1 would not lead to jamming.
- 4- The authors calculate the length to reach fully developed flow ignoring particle migration. However, the concentration is not uniform, as stated above, and the length it takes to become fully developed needs to be considered.
- 5- The authors estimate an area of influence of the particles on the wall considering a geometrical projection that completely ignores the hydrodynamics of the problem. The area on which a particle affects the flow could be significantly larger.

Board Member:

Comments to Author(s):

One of the reviewers mentions that the authors seem to have missed a significant both of literature on shear thickening shear jamming of particulate suspensions (the paper by Wyart and Cates for instance).

Even particles in a Newtonian carrier fluid tend to lead to shear thickening, but the thickening arises due to the particles (not due to the carrier fluid).

Which tendency (thickening or thinning) dominates when particles are added to a shear thinning carrier fluid is a priori unclear.

In this respect, as the reviewer mentions, there needs to be a clear conclusion about the specific role of the non-Newtonian medium – how does it create differences compared to Newtonian medium?

Another reviewer mentioned that as well as discussing rheology, there needed to be more of a review on additive manufacturing more generally. It was also suggested that thermo-effects on rheology should be discussed. This particular reviewer has included (along with their review) a version of the manuscript with the location of their various queries flagged up.

A third reviewer (like the first reviewer) also mentioned that the role of jamming needs to be considered: the role of particle concentration on jamming is crucial here. Other suspension mechanics effects (non-uniformity and/or particle migration and also areas over which the particles affect the flow) are also relevant to discuss. The weaknesses that the third reviewer flagged up mostly concerned the the modelling aspects. If the suspension modelling aspects cannot be readily fixed, the reviewer suggested that the experimental aspects may be of interest in a specialist technical journal (as opposed to an interdisciplinary journal like Proc. Roy. Soc. A)

Board member pre-assessment comments (if available):

These authors look at particulate suspensions in non-Newtonian base fluids, systems that are becoming increasingly important in additive manufacturing.

The suspensions considered involve glass beads in silicone and/or corn syrup, although the authors recognize that in manufacturing application the fluids are more likely to involve polymers.

Flow takes place in narrow capillaries that might not necessarily be orders of magnitude larger than the particles themselves.

The authors observe a power law model for the rheology of the system.

Power law fluids are of course well known in the literature (and are not themselves novel), but the authors propose relations for how the parameters of a power law model depend on solids fraction and on the ratio of the size of a capillary to the size of a particle: results like these could potentially be of use to the additive manufacturing community.

Comments:

At the end of the introduction at the top of page 4, the reader ought to be told what the rest of the manuscript covers. We are told what the objective is, but not what the manuscript will cover section by section in order to attain that objective.

The same applies to the start of section 4. It is a very long section (accounting for almost half the manuscript) but the reader starting the section has no clear idea from the outset what direction it will take and it will cover.

If the authors wish to use symbols like ω_{pic} and ω_{jc} in the conclusions they ought to remind the reader in words what they represent physically.

Author's Response to Decision Letter for (RSPA-2021-0615.R0)

See Appendix A.

RSPA-2021-0615.R1 (Revision)

Review form: Referee 1

Is the manuscript an original and important contribution to its field?

Good

Is the paper of sufficient general interest?

Good

Is the overall quality of the paper suitable?

Good

Can the paper be shortened without overall detriment to the main message?

Yes

Do you think some of the material would be more appropriate as an electronic appendix?

No

Do you have any ethical concerns with this paper?

No

Recommendation?

Accept as is

Comments to the Author(s)

The authors have clearly made significant efforts to respond to all referees' comments and improve the manuscript including widening its context and making clearer its scope. I therefore recommend it should now be published.

Review form: Referee 2

Is the manuscript an original and important contribution to its field?

Good

Is the paper of sufficient general interest?

Good

Is the overall quality of the paper suitable?

Good

Can the paper be shortened without overall detriment to the main message?

Yes

Do you think some of the material would be more appropriate as an electronic appendix?

No

Do you have any ethical concerns with this paper?

No

Recommendation?

Accept as is

Comments to the Author(s)

The authors have done a great job as presented in the manuscript. Rheology properties of the feedstock melt is a crucial piece of information for parameter settings in material extrusion additive manufacturing. I believe the publication of this manuscript can be a benefit to the AM community.

Review form: Referee 3

Is the manuscript an original and important contribution to its field?

Good

Is the paper of sufficient general interest?

Good

Is the overall quality of the paper suitable?

Good

Can the paper be shortened without overall detriment to the main message?

Yes

Do you think some of the material would be more appropriate as an electronic appendix?

No

Do you have any ethical concerns with this paper?

No

Recommendation?

Accept with minor revision (please list in comments)

Comments to the Author(s)

The authors have addressed most of my comments and concerns. I list them below with my comments.

1. I am not completely convinced that the distinction between free-flow and 'particle-interaction' regimes can be done unless the concentration is fixed.
2. The authors fully addressed my concern about the distribution of particles in the cross-section
3. Similar to comment 1. The discussion about jamming vs. 'w' would be fine if it is clear that the concentration is fixed as 'w' is varied.
4. The authors fully addressed this point
5. Here I disagree with the authors. The authors estimate the area of the wall affected by the presence of a particle geometrically, with a radial projection of the particle on the wall. However, depending on Reynolds number, the flow disturbances generated by a particle have a long range and their effect on the wall is not related to the projected area. Maybe the intersection of a

spherical shell around the particle and the tube wall could be a suitable approximation. This needs to be clarified. In other words, the impacted area should be justified from an hydrodynamic perspective.

The fact that the projected area becomes larger as the particle moves away from the tube wall, which corresponds to a larger impact area, is counterintuitive, and needs some justification.

Decision letter (RSPA-2021-0615.R1)

31-Mar-2022

Dear Dr Xia,

On behalf of the Editor, I am pleased to inform you that your Manuscript RSPA-2021-0615.R1 entitled "Rheology of Particulate Suspensions with Non-Newtonian Fluids in Capillaries" has been accepted for publication subject to minor revisions in Proceedings A. Please find the referees' comments below.

The reviewer(s) have recommended publication, but also suggest some minor revisions to your manuscript. Therefore, I invite you to respond to the reviewer(s)' comments and revise your manuscript. Please note that we have a strict upper limit of 28 pages for each paper. Please endeavour to incorporate any revisions while keeping the paper within journal limits. Please note that page charges are made on all papers longer than 20 pages. If you cannot pay these charges you must reduce your paper to 20 pages before submitting your revision. Your paper has been ESTIMATED to be 22 pages. We cannot proceed with typesetting your paper without your agreement to meet page charges in full should the paper exceed 20 pages when typeset. If you have any questions, please do get in touch.

It is a condition of publication that you submit the revised version of your manuscript within 7 days. If you do not think you will be able to meet this date please let me know in advance of the due date.

To revise your manuscript, log into <https://mc.manuscriptcentral.com/prsa> and enter your Author Centre, where you will find your manuscript title listed under "Manuscripts with Decisions." Under "Actions," click on "Create a Revision." Your manuscript number has been appended to denote a revision.

You will be unable to make your revisions on the originally submitted version of the manuscript. Instead, revise your manuscript and upload a new version through your Author Centre.

When submitting your revised manuscript, you will be able to respond to the comments made by the referee(s) and upload a file "Response to Referees" in Step 1: "View and Respond to Decision Letter". Please provide a point-by-point response to the comments raised by the reviewers and the editor(s). A thorough response to these points will help us to assess your revision quickly. You can also upload a 'tracked changes' version either as part of the 'Response to reviews' or as a 'Main document'.

IMPORTANT: Your original files are available to you when you upload your revised manuscript. Please delete any redundant files before completing the submission process.

When uploading your revised files, please make sure that you include the following as we cannot proceed without these:

- 1) A text file of the manuscript (doc, txt, rtf or tex), including the references, tables (including captions) and figure captions. Please remove any tracked changes from the text before submission. PDF files are not an accepted format for the "Main Document".
- 2) A separate electronic file of each figure (tif, eps or print-quality pdf preferred). The format should be produced directly from original creation package, or original software format.
- 3) Electronic Supplementary Material (ESM): all supplementary materials accompanying an accepted article will be treated as in their final form. Note that the Royal Society will not edit or typeset supplementary material and it will be hosted as provided. Please ensure that the supplementary material includes the paper details where possible (authors, article title, journal name). Supplementary files will be published alongside the paper on the journal website and posted on the online figshare repository (<https://figshare.com>). The heading and legend provided for each supplementary file during the submission process will be used to create the figshare page, so please ensure these are accurate and informative so that your files can be found in searches. Files on figshare will be made available approximately one week before the accompanying article so that the supplementary material can be attributed a unique DOI. Alternatively you may upload a zip folder containing all source files for your manuscript as described above with a PDF as your "Main Document". This should be the full paper as it appears when compiled from the individual files supplied in the zip folder.

Article Funder

Please ensure you fill in the Article Funder question on page 2 to ensure the correct data is collected for FundRef (<http://www.crossref.org/fundref/>).

Media summary

Please ensure you include a short non-technical summary (up to 100 words) of the key findings/importance of your paper. This will be used for to promote your work and marketing purposes (e.g. press releases). The summary should be prepared using the following guidelines:

- *Write simple English: this is intended for the general public. Please explain any essential technical terms in a short and simple manner.
- *Describe (a) the study (b) its key findings and (c) its implications.
- *State why this work is newsworthy, be concise and do not overstate (true 'breakthroughs' are a rarity).
- *Ensure that you include valid contact details for the lead author (institutional address, email address, telephone number).

Cover images

We welcome submissions of images for possible use on the cover of Proceedings A. Images should be square in dimension and please ensure that you obtain all relevant copyright permissions before submitting the image to us. If you would like to submit an image for consideration please send your image to proceedingsa@royalsociety.org

Open Access

You are invited to opt for open access, our author pays publishing model. Payment of open access fees will enable your article to be made freely available via the Royal Society website as soon as it is ready for publication. For more information about open access please visit <https://royalsociety.org/journals/authors/open-access/>. The open access fee for this journal is £1700/\$2380/€2040 per article. VAT will be charged where applicable. Please note that if the corresponding author is at an institution that is part of a Read and Publishing deal you are required to select this option. See <https://royalsociety.org/journals/librarians/purchasing/read-and-publish/read-publish-agreements/> for further details.

Once again, thank you for submitting your manuscript to Proceedings A and I look forward to receiving your revision. If you have any questions at all, please do not hesitate to get in touch.

Best wishes
Raminder Shergill
proceedingsa@royalsociety.org
Proceedings A

on behalf of
Dr Paul Grassia
Board Member
Proceedings A

Reviewer(s)' Comments to Author:

Referee: 1

Comments to the Author(s)

The authors have clearly made significant efforts to respond to all referees' comments and improve the manuscript including widening its context and making clearer its scope. I therefore recommend it should now be published.

Referee: 2

Comments to the Author(s)

The authors have done a great job as presented in the manuscript. Rheology properties of the feedstock melt is a crucial piece of information for parameter settings in material extrusion additive manufacturing. I believe the publication of this manuscript can be a benefit to the AM community.

Referee: 3

Comments to the Author(s)

The authors have addressed most of my comments and concerns. I list them below with my comments.

1. I am not completely convinced that the distinction between free-flow and 'particle-interaction' regimes can be done unless the concentration is fixed.
2. The authors fully addressed my concern about the distribution of particles in the cross-section
3. Similar to comment 1. The discussion about jamming vs. 'w' would be fine if it is clear that the concentration is fixed as 'w' is varied.
4. The authors fully addressed this point
5. Here I disagree with the authors. The authors estimate the area of the wall affected by the presence of a particle geometrically, with a radial projection of the particle on the wall. However,

depending on Reynolds number, the flow disturbances generated by a particle have a long range and their effect on the wall is not related to the projected area. Maybe the intersection of a spherical shell around the particle and the tube wall could be a suitable approximation. This needs to be clarified. In other words, the impacted area should be justified from an hydrodynamic perspective.

The fact that the projected area becomes larger as the particle moves away from the tube wall, which corresponds to a larger impact area, is counterintuitive, and needs some justification.

Board Member

Comments to Author(s):

In this revised version, the reviewers are generally supportive of the manuscript.

One reviewer has raised some issues in respect of the free flow and particle interaction regimes and also issues with jamming, capillary-to-particle size ratio (denoted ω) and the role of fixed concentration.

Another point that this same reviewer has raised is about the area of the wall affected by a particle, how this behaves with respect to distance from the wall, and also the role of flow disturbances.

It is recommended that the authors reconsider those issues, prepare a new version of the manuscript (preferably with the latest round of changes highlighted in colour) and also prepare a suitable response letter addressing the issues raised by the reviewer.

Author's Response to Decision Letter for (RSPA-2021-0615.R1)

See Appendix C.

Decision letter (RSPA-2021-0615.R2)

22-Apr-2022

Dear Dr Xia

I am pleased to inform you that your manuscript entitled "Rheology of Particulate Suspensions with Non-Newtonian Fluids in Capillaries" has been accepted in its final form for publication in Proceedings A.

Our Production Office will be in contact with you in due course. You can expect to receive a proof of your article soon. Please contact the office to let us know if you are likely to be away from e-mail in the near future. If you do not notify us and comments are not received within 5 days of sending the proof, we may publish the paper as it stands.

As a reminder, you have provided the following 'Data accessibility statement' (if applicable). Please remember to make any data sets live prior to publication, and update any links as needed when you receive a proof to check. It is good practice to also add data sets to your reference list.

Statement (if applicable): The processed experimental data is available on the SMU research archive, SMU Scholar (scholar.smu.edu).

Under the terms of our licence to publish you may post the author generated postprint (ie. your accepted version not the final typeset version) of your manuscript at any time and this can be made freely available. Postprints can be deposited on a personal or institutional website, or a recognised server/repository. Please note however, that the reporting of postprints is subject to a media embargo, and that the status the manuscript should be made clear. Upon publication of the definitive version on the publisher's site, full details and a link should be added.

You can cite the article in advance of publication using its DOI. The DOI will take the form: 10.1098/rspa.XXXX.YYYY, where XXXX and YYYY are the last 8 digits of your manuscript number (eg. if your manuscript number is RSPA-2017-1234 the DOI would be 10.1098/rspa.2017.1234).

For tips on promoting your accepted paper see our blog post:
<https://royalsociety.org/blog/2020/07/promoting-your-latest-paper-and-tracking-your-results/>

On behalf of the Editor of Proceedings A, we look forward to your continued contributions to the Journal.

Sincerely,
Raminder Shergill
proceedingsa@royalsociety.org

on behalf of
Dr Paul Grassia
Board Member
Proceedings A

Appendix A

This paper provides a systematic discussion on the shear thinning viscosity of capillary extrusion melt, which directly can be applied to tune processing parameters in extrusion based additive manufacturing. The measured viscosity data are fitted through classic power law model and the relations between the particle volume fraction and aspect ratio to the consistency index and power index are shown in detail. The results are applied in analyzing the jamming issues in extrusion based AMs. I think this paper is well written and finely prepared. I recommend the editor to let authors considering the following suggestions before accepting their manuscript for publication:

- 1) This is a rheology study tying to additive manufacturing. While the introduction provides a good review on polymer rheology models, a comprehensive review on literatures discussing the role of rheology in extrusion deposition additive manufacturing is missing. This would not be hard to accomplish as significant amount of such studies has been presented, e.g., Mackay M E. J of Rheo, 2018, Wang and Smith, Journal of composite sci. 2018, Sanchez, et al, Int J Adv Manuf Technol 2019, Das et al. ACS applied poly mat. 2021, which can be easily found through a basic google scholar search “rheology effects extrusion based additive manufacturing”.
- 2) The material model of silicone is pretty good choice, as more and more AM people are searching for innovative materials over boring PLA, ABS... Nevertheless, would it be possible that the authors could provide any literature that works with silicone AMs (Silicone gel 3DP for bio applications?). This may help amplify the important and usefulness of your work.
- 3) I must have been missing something. But in Figure 2, it is hard to determine where is “L1” as well as “L2”. Please clarify it. In addition, are these two lengths are given implicitly? I did not found explicit information after this figure, neither.
- 4) Page 6-Line 48: τ_w and $\dot{\gamma}_w$ seem to be different from what appear in Eqn. (2.5), is that on purpose? Please explain.
- 5) Page 6-Line 50: To quantify the impacts of xxx, xxx, xxx on capillary melt rheology (please change if this statement is not suitable), xxx and xxx were obtained ...
- 6) Page 8- Line 25: the highlighted “a” in “a may depend on the suspension fluid” shall be revised in the same format as appeared in Eqn. (4.1)
- 7) Page 10-Line 40: previously, the paper states that α may depend on the suspension fluid. While here, the paper says n is a function of ϕ , exclusively. If so, it is recommended to define that the adjust parameter α is a constant and/or depends on ϕ .
- 8) Page 18-Line9: why does thermo-effects greatly influence the rheology of polymer melts in AM? There are papers out there discussing this issue (e.g., Ouyang, et al Physics of Fluids, 2020, 32(5)). As a paper designated for extrusion based AM, I recommend the authors provide a little further discussion on it. This may be an interesting direction for future works of this work.

Appendix B

We would like to thank the reviewers for their careful review of the manuscript and for providing helpful suggestions and comments that have served to improve the revised manuscript. We have provided point-by-point responses to the comments from each reviewer below, with our responses to each comment indented to distinguish them from the rest of the text. The page and line numbers in the responses are based on the revised manuscript PDF file.

Referee 1:

1. [page 3] Line 52 – mention that much previous work looks at shear thickening ‘material’ – however it is important here to realize that thickening is usually the result of the particulate phase, rather than a particulate phase in a thickening medium. Given the potentially separate roles of medium rheology and the particulate phase care should be taken to be clear on what is thickening, what is thinning.

This is a good point. This text (now on the bottom of p. 3 and top of p. 4) has been updated to clarify the meaning of shear thickening material. The updated text is below:

“Additionally, most of the work utilizes particle-induced shear thickening material (particle suspensions in a Newtonian fluid that become shear thickening after adding particles) [28][30][31] or viscoelastic materials in a squeeze geometry [32], but most of the polymer materials used in AM are shear thinning.”

2. On this point, the choice of literature referred to regarding the ‘jamming’ role of the particle phase is a bit eclectic – there are lots of more recent works by Wyart, Cates, etc representing probably the most recent work on high concentration particle suspension rheology (albeit generally in Newtonian fluids) and the relevance of this work ought to be considered in the paper.

Thank you for the additional suggested references. The discussion of particle-phase jamming in the introduction has been rewritten to incorporate these and additional references regarding jamming. The updated paragraphs (starting with the 5th full paragraph on p. 3) are included below:

“The observations of suspensions in both Newtonian and non-Newtonian fluids relate that with higher ϕ , the materials change from a fluid-like to a solid-like state with an observable yield stress, which is a phase change called the jamming transition [23]. The jamming transition occurs at a characteristic volume fraction (ϕ_M) [24] which in general may depend on the nature of the particles and the flow state (e.g., shear rate) of the suspension [25][26][27]. As $\phi \rightarrow \phi_M$, lubrication layers between particles begin disappearing and the number of frictional contacts per particle increases [25]. At ϕ_M the suspension reaches the “maximum packing fraction possible for a given suspension composition and packing arrangement” [24].

Much of the work on jamming behavior focuses on modeling the mechanics of the jamming process and on determining the jamming transition utilizing generic shear flows [25][26][27][28][29]. In AM applications, a significant factor impacting jamming includes confinement effects from the capillary inner diameter (D) in relation to the particle mean diameter (d) and its distribution, but this has not been considered to date. Additionally, most of the work utilizes particle-induced shear thickening material (particle suspensions in a Newtonian fluid that become shear thickening after adding particles) [28][30][31] or viscoelastic materials in a squeeze geometry [32], but most of the polymer materials used in AM are shear thinning.”

Additionally, the discussion of ϕ_M on p. 17 has been updated with additional references. However, a detailed assessment of the evolution of particle contacts as jamming is approached following the model of Wyart and Cates (2014) is beyond the scope of the present work.

3. Was there any evidence of issues with the contraction from syringe into capillary especially at high particle fraction? Previous work (e.g. Isa, Haw etc) has shown significant impacts of contraction and greater degree of ‘jamming’ due to the extensional flow. It is somewhat surprising this is not observed at 40% fraction for example and it would be good to have a clear confirmation that this is the case. One of the issues with such jamming is that the fluid filters through the jammed particles (eg Haw 2004) and so the downstream volume fraction is decreased.

Thank you for pointing out this potential concern. “Self-filtration” due to jamming, as described by Haw (2004) is not something that was observed in these experiments. To clarify this point and suggest some reasons for the difference with respect to the results of Haw (2004), the following paragraph was added to the end of section 3 (p. 6 bottom and p. 7 top), after “corresponding to the approach to the jamming transition as the capillary diameter approached the particle mean diameter”:

“Despite the flow contraction entering the capillary tube in the rheometer (see Figure 2), there did not appear to be self-filtration due to intermittent jamming at the contraction as reported in Haw [39]. For example, UV 225-1 suspensions with A3000 particles for $\phi = 30\%$ at $\omega = 43.0$ and a volume flow rate of 0.1 mL/min gave mass flow rates of 0.150 ± 0.001 g/min leaving the capillary tube. Based on the density of the suspension fluid and particles, the expected mass flow rate was 0.150 g/min, indicating the suspension fluid and particles were exiting the capillary in the same volume ratio at which they entered because the density of the particles was approximately 2.5 times larger than that of the fluid. The reason no self-filtration was observed could be because the critical ϕ to observe this behavior was not reached, or because of differences in the suspension fluid, which in this case was both shear thinning and $\sim 10^4$ times more viscous than the suspension fluid used in Haw [39]. Likely, a combination of these factors was at play.”

4. Overall there is not much clear conclusion about the specific role of non-Newtonian medium – how does it create differences compared to Newtonian medium? This seems to be the point of the paper at the start but it is not clearly concluded on. This ought to be improved to help readers grasp the key outcome even if they are not going to follow the details of the models presented.

The results indicated the dominant effect of ω on the rheology, so the discussion focused on this parameter. Nevertheless, the distinction from the case with a Newtonian medium is discussed in section 4 (a), where the flow behavior index results are presented. To make it clearer, the following statement was added before equation (4.1) (p. 7) in section 4(a), after the sentence ending “which is also the case for all the models in Table 1.”:

“That is, over the range of conditions studied, for a suspension of particles in a Newtonian suspension fluid at fixed ϕ , the suspension remained Newtonian, whereas the suspension became more shear thinning when particles were added to a shear thinning suspension fluid.”

Additionally, the second paragraph of the Conclusion (section 5, p. 18) was updated to the following:

“Based on the experimental results, the flow behavior index n was found to be only dependent on ϕ for a suspension with the same suspension fluid. A linear relation between n and ϕ was observed and modeled with an empirical equation with acceptable accuracy. The primary difference observed between suspensions with Newtonian and shear-thinning suspension fluids was that adding particles to a shear-thinning suspension fluid made the suspension more shear thinning while a suspension in a Newtonian suspension fluid remained Newtonian.”

Referee 2:

1. This is a rheology study trying to additive manufacturing. While the introduction provides a good review on polymer rheology models, a comprehensive review on literatures discussing the role of rheology in extrusion deposition additive manufacturing is missing. This would not be hard to accomplish as significant amount of such studies has been presented, e.g., Mackay M E. J of Rheo, 2018, Wang and Smith, Journal of composite sci. 2018, Sanchez, et al, Int J Adv Manuf Technol 2019, Das et al. ACS applied poly mat. 2021, which can be easily found through a basic google scholar search “rheology effects extrusion based additive manufacturing”.

Thank you for the suggested references. The following updates have been made to the introduction:

Page 2, line 9 of the first paragraph of the Introduction:

“Hence, the rheology of the suspension governs its extrusion behavior, which is important for extrusion deposition in AM [2][3].”

The end of the next to last paragraph in the introduction has been rewritten as a new paragraph (top of p. 4) and includes additional information on additive manufacturing:

“Prior work in AM has also considered the flow behavior of particulate composites. Some studies have investigated the rheology of carbon fiber (CF) reinforced polymer composites showing general shear thinning behavior and that the addition of CF can increase the shear thinning characteristics of the polymer melt [33][34], but the range of ϕ investigated was limited and confinement effects were not considered. Wang and Smith [35], [36] used computational methods to simulate the flow behavior of fiber-based polymer composites inside a nozzle, including model rheology effects. The emphasis in these studies, however, was on fiber orientation following printing and the resulting mechanical properties of the solid printed material.”

2. The material model of silicone is pretty good choice, as more and more AM people are searching for innovative materials over boring PLA, ABS... Nevertheless, would it be possible that the authors could provide any literature that works with silicone AMs (Silicone gel 3DP for bio applications?). This may help amplify the important and usefulness of your work.

Silicone was used in this investigation instead of a thermoplastic material as it has similar rheology (shear thinning) but doesn't require heating to achieve a liquid state. The silicone was not intended as a substitute material for AM, but rather as an analog material to understand rheology under conditions relevant for AM. So, we don't feel it is necessary to discuss the silicone gel printing here.

3. I must have been missing something. But in Figure 2, it is hard to determine where is “L1” as well as “L2”. Please clarify it. In addition, are these two lengths are given implicitly? I did not found explicit information after this figure, neither.

Thank you for catching this error. This text was supposed be in the explanation of Figure 3 instead of Figure 2. This has been updated.

4. Page 6-Line 48: τ_w and $\dot{\gamma}_w$ seem to be different from what appear in Eqn. (2.5), is that on purpose? Please explain.

Sorry for the typo. This has been updated to τ and $\dot{\gamma}$ in the revised text on p. 6 line 5.

5. Page 6-Line 50: To quantify the impacts of xxx, xxx, xxx on capillary melt rheology (please change if this statement is not suitable), xxx and xxx were obtained.

The statement “on the particulate suspension rheology in the capillary, the” was added to clarify. See the last paragraph in section 2 (p. 6).

6. Page 8- Line 25: the highlighted “a” in “a may depend on the suspension fluid” shall be revised in the same format as appeared in Eqn. (4.1)

Sorry for the typo. This has been updated.

7. Page 10-Line 40: previously, the paper states that α may depend on the suspension fluid. While here, the paper says n is a function of ϕ , exclusively. If so, it is recommended to define that the adjust parameter α is a constant and/or depends on ϕ .

Thank you for pointing out this inconsistency. This statement has been and updated to the following: “Since n with the same suspension fluid is a function of ϕ only as discussed above...” (see p.11 in the paragraph preceding equation (4.8)).

8. Page 18-Line9: why does thermo-effects greatly influence the rheology of polymer melts in AM? There are papers out there discussing this issue (e.g., Ouyang, et al Physics of Fluids, 2020, 32(5)). As a paper designated for extrusion based AM, I recommend the authors provide a little further discussion on it. This may be an interesting direction for future works of this work.

The intent of this statement was simply to convey the well-known importance of temperature on the rheology of polymer melts (i.e., that viscosity decreases as temperature increases), and that this factor was not considered in the present study. Additional references were added following this statement to help clarify the meaning (see the last paragraph of the conclusion on p. 19). As noted in the conclusion, it is definitely a point worthy of further study.

Referee 3:

1. The authors separate free-flow and particle-interaction regions depending on the particle-to-capillary size ratio 'w'. This is probably incorrect. At high concentrations there should be particle-particle interactions independent of the value of 'w'.

We agree that there will be particle-particle interactions even under “free-flow” conditions. The term “free-flow” was not meant to indicate the particles don’t influence each other, but rather that conditions here are dominated by an intervening lubrication layer of suspension fluid between the particles that allows for smooth suspension flow for the range of ϕ investigated, while the “particle interaction” region is dominated by increasing particle-to-particle and particle-to-wall frictional contacts as ω is decreased, leading ultimately to jamming. The description of these regions in the second paragraph of Section 4(b) (p. 8) has been updated to clarify this. The updated text is below:

“The general trend observed for K is illustrated schematically in Figure 6. The minimum in K occurs at a boundary defined by ω_{pic} , below which the rheology is dominated by increasing particle-to-particle and particle-to-wall frictional contacts as ω is decreased until sufficient contacts per particle are achieved to effect jamming at ω_{jc} . The “particle interaction” condition governs the flow for $\omega_{jc} < \omega < \omega_{pic}$, due to the strong contact interactions in this region. Conversely, the “free flow” condition occurs for $\omega > \omega_{pic}$, where the rheology is dominated by an intervening lubrication layer of suspension fluid between the particles that allows for smooth suspension flow for the range of ϕ tested ($\phi < \phi_M$). This region includes both the constant K behavior as $\omega \rightarrow \infty$ and the dip in K as ω decreases toward ω_{pic} . The decrease in K is not treated as separate behavior as measurements [1] (discussed below) showed that in this region the particles were still uniformly distributed in the capillary cross section similar to larger ω behavior.”

2. The authors claim that experiments have shown that the distribution of particles is uniform in the cross section for large 'w' values. This is incorrect. There is a large body of work on suspension flows, including several theoretical models, describing the non-uniform distribution of particles in the cross section.

We agree there is previous work showing a non-uniform distribution of particles in the cross-section of suspension flows in tubes. Nevertheless, that was not observed in the present experiments. This is likely related to the differences in flow geometry and fluid properties compared to prior studies. For extremely long tubes, a non-uniform particle distribution may appear with the materials used in this study, but such behavior would be unrealistic for the target application of additive manufacturing (where flow paths tend to be relatively short). As the particle distribution was observed to be uniform throughout the tubes for the conditions investigated, this assumption was used in the modeling. The following paragraphs and data were added to the end of Section 4(b) (p. 9) to clarify this point:

“The observed uniformity of the particle distribution in the capillary cross section deserves further discussion. Generally, the inhomogeneous shear experienced by the particles in these flows is known to induce a non-uniform particle distribution sufficiently far downstream in the tubes, even for shear-thinning suspension fluids [24][42][43]. This behavior, however, was not observed in the present investigation. To confirm this, particle distributions inside the capillary at the entrance and (hydrodynamically) fully-developed region were visualized via a micro-CT (SkyScan 1172 high-resolution desktop scanner at a resolution of 2.00 μm). For these measurements, samples were fabricated by extruding the suspension (with the addition of a curing agent) into transparent tubing ($D = 1.286\text{mm}$) and then curing the silicone polymer with a UV light to create a solid sample that could be further analyzed. Three samples were fabricated including samples cured during and after extrusion in the (hydrodynamically) fully-developed region, and a sample cured after extrusion in the entrance region. A sample cross-section obtained from the micro-CT scans is shown in Figure 7 (a) for $\phi = 10\%$ (UV 225-1 with A3000 particles). The lighter regions are the particles and the darker regions are the silicone. Hence, the radial particle density distribution can be obtained by calculating the average of grayscale of the sample at different radial positions for all axial cross sections in a given region. The result for this sample is shown in Figure 7 (b), which shows the particle volume fraction is uniform in the tube cross-section from the entrance to the fully-developed region. Similar results were obtained for $\phi = 30\%$.

This difference between the present results and other studies showing particle migration and non-uniform particle distributions in particulate suspensions extruded through tubes is likely related to the differences in flow geometry and fluid properties. Tehrani [42] indicates the radial migration velocity (V_r) of in non-Newtonian fluids is determined by

$$V_r \propto d^2 We \frac{\partial \dot{\gamma}}{\partial r}$$

where We is the Weissenberg number given by the ratio of the first normal stress difference to the shear stress (N_1/τ). Then the entrance length for the particle distribution scales like $L_{ep} \sim UD/V_r$ where U is the average velocity in the tube. Using $\dot{\gamma} \sim U/D$ and $\partial \dot{\gamma} / \partial r \sim \dot{\gamma} / D$ gives

$$\frac{L_{ep}}{D} \sim \frac{\omega^2}{We}$$

Considering a typical shear rate of 50 s^{-1} [225-1] were in the range $2.2 \sim 36$ [42] $We \approx 0.47$ [225-1] $\omega > 30$ [42] $L_{ep}/D \gtrsim 10^2$ that in Tehrani [42], leading to $L_{ep}/D > 1000$ for the present results. Higher ϕ ($\phi = 10\%$ and 12%), but increasing ϕ investigated here more shear thinning (as noted above), which can produce more plug-like flow and slow particle migration (increase L_{ep}). So, while measurable particle migration might appear for extremely long capillary lengths, such behavior would be unrealistic for the target application of additive manufacturing (where flow paths tend to

be relatively short) and uniform particle distributions will be assumed in the following. long capillary lengths, such behavior would be unrealistic for the target application of additive manufacturing (where flow paths tend to be relatively short) and uniform particle distributions will be assumed in the following.

Figure 7. (a) Cross section of a micro-CT scan of a particulate suspension ($\phi = 10\%$), (b) Average gray scale of micro-CT scans vs. radial position ($\phi = 10\%$).

Using the assumption of uniform particle distribution in the free flow condition, mathematical models for the observed trends in K are developed in the following based on the governing behavior (free flow vs. particle interaction) for each condition.”

Additionally, the following sentence is added after equation (4.19) on p. 13:

“(For behavior with a non-uniform particle distribution, Equation 4.19 can be modified accordingly.)”

3. The authors discuss jamming as dependent on 'w'. In general, it has to depend on the concentration as well. For low enough concentrations, even 'w' close to 1 would not lead to jamming.

We agree that jamming is also dependent on the particle concentration (ϕ) and it was presented and modeled in the paper as a function of both ϕ and ω as indicated in equations (4.29) and (4.30).

The behavior of ω_{jc} for low volume fraction is discussed at the top of p. 17, but the behavior depends on the particle size distribution as discussed in the third paragraph of the conclusion (bottom of p. 18).

4. The authors calculate the length to reach fully developed flow ignoring particle migration. However, the concentration is not uniform, as stated above, and the length it takes to become fully developed needs to be considered.

The concern about non-uniformity of the flow is discussed in the response to concern #2 above. The entrance length in the design of the experiment only considers the hydrodynamic entrance length given the uniformity of the particle distribution. Page 5, the last paragraph have been revised to clarify this point:

“Lengths of the capillaries were determined to guarantee the flow in the region L was (hydrodynamically) fully-developed and laminar based on the related theories [37]. (The particle distribution remains uniform throughout as will be discussed in Section 4(b), so no entrance length for this property was considered.)”

5. The authors estimate an area of influence of the particles on the wall considering a geometrical projection that completely ignores the hydrodynamics of the problem. They are on which a particle affects the flow could be significantly larger.

The assumption is made based on the projection and the distance between the particle and the capillary. The hydrodynamics comes into the area of influence result in equation (4.16) via the parameter q_i (see equations (4.11) through (4.14)). We agree the estimated area of influence is only approximate, but it should scale according to equation (4.16), and the empirical parameters included in the model provide flexibility to correct for inaccuracy in this result.

Additional Board Member Comments

1. At the end of the introduction at the top of page 4, the reader ought to be told what the rest of the manuscript covers. We are told what the objective is, but not what the manuscript will cover section by section in order to attain that objective. The same applies to the start of section 4. It is a very long section (accounting for almost half the manuscript) but the reader starting the section has no clear idea from the outset what direction it will take and it will cover.

The following has been added to the beginning of Section 4 (p. 7):

“In this section, models describing the behavior of K and n in terms of ω and ϕ will be developed.”

Additionally, an outline of the manuscript contents has been added to the end of the introduction (p. 4):

“In this work, Section 2 presents the design of the rheometer used to investigate the particulate suspensions and the range of conditions investigated. Section 3 presents the measured rheology of the suspensions in terms of the flow consistency index (K) and flow behavior index (n) for the observed power-law behavior of the suspensions. Section 4 develops models describing measured trends in K and n . Section 5 presents the conclusions.

2. If the authors wish to use symbols like ω_{pic} and ω_{jc} in the conclusions they ought to remind the reader in words what they represent physically.

Symbols are reemphasized the first time they are used in the conclusion as suggested.

3. Based on the revisions made to the manuscript, the sentences leading to equation (4.29) on p. 17 were updated as follows:

“The value of ϕ_M may be impacted by many factors, including the flow state and polydispersity [41][25][26]. To simplify the analysis, it will be assumed that $\phi_M \approx 0.585$, which is appropriate for monodisperse spheres [27] and is close to the range of values for bidisperse spheres [26]”

The change in the value of ϕ_M used in the model changed C_4 and b_4 to $C_4 = 0.55 \pm 0.16$ and $b_4 = 1.62 \pm 0.28$, but the behavior of the model and conclusions did not change.

Appendix C

We would like to thank the reviewers again for their careful review of the manuscript. The remaining concerns from Referee 3 are addressed point-by-point below. The response to each concern is provided below the concern and indented.

Referee 3:

The authors have addressed most of my comments and concerns. I list them below with my comments.

1. I am not completely convinced that the distinction between free-flow and 'particle-interaction' regimes can be done unless the concentration is fixed.

This description was intended for the situation with fixed concentration. For clarity, we have updated the first sentence of the second paragraph of Section 4(b) to “The general trend observed for K at fixed ϕ is illustrated schematically in Figure 6.” We also updated the caption for Figure 6 to “Categorization of different flow conditions observed for the dependence of K on ω at fixed ϕ .”

2. The authors fully addressed my concern about the distribution of particles in the cross-section

Thank you. We are glad the revised manuscript has addressed this point.

3. Similar to comment 1. The discussion about jamming vs. 'w' would be fine if it is clear that the concentration is fixed as 'w' is varied.

The updates to Section 4(b), noted above, apply to all of section 4. However, to clarify this point, we have updated the discussion of jamming in the second paragraph after “Model for ω_{jc} and b ” on p. 16. The first sentence of this paragraph has been changed to:

“According to the experimental results, for constant ϕ jamming occurs at larger ω as ϕ is increased.”

Also, the phrase before equation (4.29) was changed to

“To capture the dependence of ω_{jc} on ϕ (for a fixed ϕ), it is represented as”

4. The authors fully addressed this point

Thank you. We are glad the revised manuscript has addressed this point.

5. Here I disagree with the authors. The authors estimate the area of the wall affected by the presence of a particle geometrically, with a radial projection of the particle on the wall. However, depending on Reynolds number, the flow disturbances generated by a particle have a long range and their effect on the wall is not related to the projected area. Maybe the intersection of a spherical shell around the particle and the tube wall could be a suitable approximation. This needs to be clarified. In other

words, the impacted area should be justified from an hydrodynamic perspective.

The fact that the projected area becomes larger as the particle moves away from the tube wall, which corresponds to a larger impact area, is counterintuitive, and needs some justification.

The model for the wall area affected by the particle emphasizes the behavior for particles near the wall. Near the wall, only the area close to the particle is relevant due to the curvature of the particle. As particles move further away, the influence is less concentrated near the particle (the velocity gradients are more diffuse as the distance of the particle from the wall approaches and exceeds the particle radius) and the affected area is expanded. At greater distances, a spherical influence region would probably be more appropriate as the reviewer suggests, but the influence of the particle on the flow has diminished by this point, as is already captured by the model.

The following sentence has been added after equation (4.16) for clarification:

“This construction emphasizes that the dominant influence of the particle narrows to the region just between the particle and the wall as particles approach the wall, where the effect on the wall shear stress is greatest.”

Reynolds number is not explicitly accounted for as the effect of Reynolds number is expected to be weak over the range of Reynolds numbers considered for these very viscous suspensions.

Additional changes to save space and shorten the paper:

- a) Figures 10 and 11 were merged into Figure 10.
- b) Figure 14 was eliminated and some text was added to the preceding paragraph to describe the effects illustrated in the figure.